# Integrated mRNA and miRNA Transcriptome Analysis Suggests a Regulatory Network for UV–B-Controlled Terpenoid Synthesis in Fragrant Woodfern (*Dryopteris fragrans*)

**DOI:** 10.3390/ijms23105708

**Published:** 2022-05-20

**Authors:** Chunhua Song, Yalin Guan, Dongrui Zhang, Xun Tang, Ying Chang

**Affiliations:** 1College of Life Sciences, Northeast Agricultural University, Harbin 150030, China; chunhuasong@163.com (C.S.); geraldtwinersom@gmail.com (D.Z.); tangxun1119@gmail.com (X.T.); 2College of Marine Life Sciences, Ocean University of China, Qingdao 266100, China; guanyalin0710@163.com

**Keywords:** Fragrant woodfern, terpenoid metabolic pathways, UV–B, mRNA, small RNA

## Abstract

Fragrant woodfern (*Dryopteris fragrans*) is a medicinal plant rich in terpenoids. Ultraviolet-B (UV–B) light could increase concentration of terpenoids. The aim of this study was to analyze how UV–B regulates the terpenoid synthesis of the molecular regulatory mechanism in fragrant woodfern. In this study, compared with the control group, the content of the terpenes was significantly higher in fragrant woodfern leaves under UV–B treatment for 4 days (d). In order to identify how UV–B regulates the terpenoid metabolic mechanism in fragrant woodfern, we examined the mRNAs and small RNAs in fragrant woodfern leaves under UV–B treatment. mRNA and miRNA–seq identified 4533 DEGs and 17 DEMs in the control group compared with fragrant woodfern leaves under UV–B treatment for 4 d. mRNA–miRNA analysis identified miRNA target gene pairs consisting of 8 DEMs and 115 miRNAs. The target genes were subjected to GO and KEGG analyses. The results showed that the target genes were mainly enriched in diterpene biosynthesis, terpenoid backbone biosynthesis, plant hormone signal transduction, MEP pathway and MVA pathway, in which miR156 and miR160 regulate these pathways by targeting *DfSPL* and *DfARF*, respectively. The mRNA and miRNA datasets identified a subset of candidate genes. It provides the theoretical basis that UV–B regulates the terpenoid synthesis of the molecular regulatory mechanism in fragrant woodfern.

## 1. Introduction

Fragrant woodfern (*Dryopteris fragrans*) is a perennial medicinal herb, mainly produced in Northeast China [1]. Fragrant woodfern has a very good effect on various skin diseases and rheumatoid arthritis, and is known as “the nemesis of skin diseases” [2]. So far, a series of compounds have been isolated from fragrant woodfern, the main types include phloroglucinols, terpenes, flavonoids, and ophenylpropanoids [3]. It has a wide range of therapeutic applications including antimicrobial, antitumor, anti-inflammatory, antioxidant, and antistress properties [4,5]. Terpenoids are the characteristic chemical components of fragrant woodfern and the main effective components that play related functions. Therefore, attempts were made to improve the terpenoid production by optimizing chemical and physical environmental factors or elicitation using abiotic factors including water deficit [6], heavy metal, [7], and wounding [8]. Previous reports have shown that Ultraviolet–B (UV–B) light could induce significant over-expression of key genes in terpenoid biosynthesis and led to an increase in the concentration of terpenoids [9]. However, the regulatory mechanism of UV–B on plant terpenoid metabolic pathways is still very limited.

Terpenoids are a group of secondary metabolites widely found in plants despite their great structural diversity. The biosynthetic pathways of all terpenoids are mainly the cytoplasmic mevalonate pathway (MVA pathway) and the 2-C-methylerythritol 4-phosphate pathway (MEP pathway) in the plastids [10]. The common precursors for both pathways to produce terpenoids are isopentenyl diphosphate (IPP) and its allylic isomer dimethylallyl diphosphate (DMAPP), respectively [11]. IPP and DMAPP are catalyzed by geranyl diphosphate synthase (GPS), farnesyl diphosphate synthase (FPS), Geranylgeranyl diphosphate synthase (GGPPS) and geranylfarnesyl diphosphate synthase (GFPPS) to form geranyl diphosphate (GPP, C10) [12], farnesyl diphosphate (FPP, C15) [13], Geranylgeranyl diphosphate (GGPP, C20) [14] and geranylfarnesyl diphosphate (GFPP, C25) [14,15] respectively, which are cyclized and modified to form different terpenoids. More and more studies have shown that miRNAs have important effects on plant growth and development [16], hormone responses [17] biotic and abiotic stresses [18], and immune responses [19]. Moreover, miRNAs play an important role in regulating the production of secondary metabolites in plants. miRNAs affect terpenoid synthesis by mediating the expression of rate-limiting enzyme genes in the terpenoid synthesis pathway. miR854e ginseng was identified as targeting *PgFPS* on the ginsenoside synthesis pathway in *Panax ginseng* [20]. Vashisht et al., identified miRNA4995 targeting 3-deoxy-7-phosphoheptulonate synthase gene, a key enzyme in the phenylpropanoid synthesis pathway, thereby affecting picroside production [21]. However, the regulatory mechanism of miRNAs involved in terpenoid metabolic pathways needs to be further explored.

UV–B radiation (280–320 nm) is an important photoecological factor, which has a significant impact on humans, animals, plants, and ecosystems [22,23,24]. At present, it is particularly noteworthy that research on the effects of UV–B radiation stress on plant growth has gradually deepened [25]. Among them, the accumulation of plant secondary metabolites (terpenoids, alkaloids, phenol, and secondary nitrogen-containing compound) is a stress response that has been studied in this field [26,27,28]. Dolzhenko et al., studied the effect of UV–B radiation on the components of *Mentha piperita*, and the results showed that UV–B radiation changed gene expression, enzyme activity, and the accumulation of defensive metabolites. The qRT–PCR analysis shows that this effect also includes the biosynthesis of terpenoids and the expression of coding genes. It is believed that under the regulation of UV–B, terpenoids and flavonoids in peppermint can be transformed into each other [29]. Previous reports have shown that ultraviolet (UV) light could induce significant overexpression of key genes in artemisinin biosynthesis and led to an increase in the concentration of artemisinin [9]. Wei Ning et al., picked fresh honeysuckle (*Lonicera japonica*) flower buds and treated them with UV–B radiation. They detected four iridoid glycosides such as loganic acid, oxidized loganin, serocyclin and (E)-aldosecologanin. Significant increases cause the antioxidant capacity to also significantly increase [30]. Moreover, after UV–B irradiation, plants display diverse morphological and physiological responses that are likely to be involved in signal transduction cascades and gene expression in the biosynthesis of secondary metabolites [31].

In order to understand the complexity of UV–B responses in fragrant woodfern and continue our work of the elicitation on terpenoid biosynthesis [32,33,34], the present work was carried out to investigate the mRNA and miRNA transcriptome, terpenoid composition and relative content, and physiological changes in fragrant woodfern to UV–B radiation. A low dose of UV–B radiation was applied to fragrant woodfern seedlings for a short term to enhance terpenoid production. mRNA and miRNA transcriptomes were employed to analyze the gene expression profiles related genes to terpenoid biosynthesis. We also observed reactive oxygen species (ROS) generation under UV–B radiation. The microarray data were validated through quantitative RT–PCR (qRT–PCR) and differentially expressing candidates were identified for further function analysis on terpenoid biosynthesis.

## 2. Results

### 2.1. Cell Death, Photosynthetic Pigments Content, ROS and Peroxidase Activity in Fragrant Woodfern Leaves after UV–B Treatment

UV–B radiation can cause the imbalance of ROS metabolism and produce a large amount of superoxide ions, which intensifies the peroxidation of membrane lipids, eventually causing damage to the membrane structure and damage to the normal physiological functions of plants. We examined cell death, photosynthetic pigments content, ROS, and peroxidase activity in fragrant woodfern leaves under UV–B treatment. As shown in the Appendix A, the 6 d treatment of UV–B on fragrant woodfern seedlings induced foliar injury and cell death estimated by Evans blue staining (Appendix A). Compared with the control group, a significant decrease in chlorophyll a and chlorophyll b content was found after 2 d of UV–B treatment (Appendix A, C; *p* < 0.05), while total carotenoid content increased significantly after 2 d (Appendix A, *p* < 0.05). Compared with the control group, significant decrease in the contents of H_2_O_2_, O_2_^−^ and MDA, and the activities of SOD, POD and CAT was found after 2 d of UV–B treatment (Appendix A; *p* < 0.05). 

### 2.2. Terpenoid Content in Fragrant Woodfern Leaves under UV–B Treatment

To investigate whether terpenoids in the leaves of fragrant woodfern are involved in the resistance response under UV–B stress, we treated fragrant woodfern with UV–B induction and collected metabolic samples of the leaves. We quantified the samples at different times of UV–B treatment by GC–MS method. As shown in the Appendix A, we identified a total of nine terpenoids, namely Pentyl filicinate, β-Humulene, α Muurolene, Isolongifolol, Aristolochene, β-Maaliene, Drimenol, β-Cadinen and α-Patchoulene (Appendix A, Appendix A). Compared with the control, the sum of the nine terpenes content did not change significantly (*p* > 0.05) at 0, 2 and 6 d of UV–B treatment, but the content of the nine terpenes was significantly higher (*p* < 0.05) at 4 d of UV–B treatment. So, we selected fragrant woodfern leaves for mRNA and miRNA-seq analysis for 6 d treatment of UV–B.

### 2.3. Overview of RNA-Seq Dynamics and Small RNA Sequencing

The raw read values of the samples ranged from 20,892,938 (CK2) to 24,839,653 (CK1); the clean read values of the samples ranged from 20,220,033 (CK2) to 20,220,033 (CK1); the clean base values of the samples ranged from 6.07 G (CK2) to 7.19 G (CK1); the error rate value of the samples was 0.02%; the Q20 values of the samples ranged from 97.66% (UV3) to 97.93% (CK1); the Q30 values of the samples ranged from 93.44% (UV3) to 94.42% (CK1); the GC content values of the samples ranged from 45.9% (CK3) to 46.92% (CK1) (Table 1); and the assembly produced a total of 141,963 transcripts. Then, tgicl software was used on the transcripts to remove abundance, and 69,493 genes were gained; the N50 statistic was 2305, which meant that more than 50% of the genes were longer than 2305 bp; the N90 statistic was 526, which meant that more than 90% of the genes were longer than 526 bp; the length distribution of all the assembled yam genes shown in Appendix A, which indicated that 27.3% of the complete transcripts and 21.01% of the total genes were longer than 2000 bp. A total of 69,493 genes were functionally annotated with seven functional database: (NR, NT, GO, KOG, KEGG, SwissProt and PFAM), 34,707 (49.94%), 9274 (13.34%), 33,663 (48.44%), 15,550 (2037%), 15,025 (21.62%), 33,479 (48.17%) and 33,668 (48.44%) reads were annotated functionally, respectively (Appendix A); 4003 genes were shared among the five functional database; the square of the Pearson correlation coefficient (R2) ranged from 0.662 to 0.815; a heat map cluster showed good correlations among the replicates which indicated high repeatability of the data; and FPKM box–plot density distribution gene expression is mainly distributed between 0.1–0.8 (Appendix A). In conclusion, these assessments showed that variability in gene expression among the replicates for the same tissues was much lower than that among the two different samples, indicative of the high quality of the dataset. 

Furthermore, the corresponding six small RNA libraries at the two samples were also constructed for deep sequencing. Initially, a total of 85,811,691 reads were generated. After removing adaptors, the low-quality reads, including reads with lengths <18 nt or >30 nt, the remaining clean reads were ranging from 11,077,797 clean reads (UV3) to 16,386,263 clean reads (UV1) with an average of 13,636,575 reads (Table 2). Subsequently, 6,307,041 (58.03%), 4,561,120 (51.72%), 6,417,540 (67.93%), 6,917,716 (61.08%), 4,580,695 (57.97%) and 4,023,081 (54.21%) reads were mapped in the sRNA database (rRNA, tRNA, snRNA, and snoRNA), respectively (Table 2). The square of the Pearson correlation coefficient (R2) ranged from 0.793 to 0.926. Data analysis showed that 8 known miRNAs and 94 novel miRNAs belonged to 10 miRNA families (Appendix A). 

### 2.4. Differentially Expressed Genes Annotation by GO Term and KEGG Pathway

In total, 4533 differentially expressed genes (DEG) were obtained in CK compared with UV–B, including 3022 up-regulated DEGs and 1511 down-regulated DEGs (Figure 1A). The detected DEGs were annotated to 2124 cellular components, 489 biological processes, and 939 molecular functions. The GO enrichment analysis revealed that 56 GO terms were significantly (*p* < 0.05) enriched. The biological process included protein phosphorylation (GO:0006468, 235), oxidation-reduction process (GO:0055114, 364), phosphorylation (GO:0016310, 286), phosphate-containing compound metabolic process (GO:0006796, 387), phosphorus metabolic process (GO:0006793, 390), cellular protein modification process (GO:0006464, 353), protein modification process (GO:0036211, 353) and response to hormone (GO:0009725, 19); the cellular components included thylakoid (GO:0009579, 49), photosystem (GO:0009521, 42), thylakoid part (GO:0044436, 48) and photosynthetic membrane (GO:0034357, 43); and the molecular function included transferase activity (GO:0016740, 705), oxidoreductase activity (GO:0016491, 373), protein kinase activity (GO:0004672, 238), kinase activity (GO:0016301, 278), ion binding (GO:0043167, 920) and catalytic activity (GO:0003824, 1527) (Figure 1C).

For KEGG enrichment, the DEGs were involved in a total of 113 metabolic pathways. The 19 enriched metabolic pathways were identified by Fisher’s exact test (*p* < 0.05), including photosynthesis (ko00195), flavonoid biosynthesis (ko00941), plant hormone signal transduction (ko04075), phenylpropanoid biosynthesis (ko00940), glutathione metabolism (ko00480), starch and sucrose metabolism (ko00500), diterpenoid biosynthesis (ko00904) and terpenoid backbone biosynthesis (ko00900) (Figure 1D). Among these GO terms and KEGG, eight key genes were identified, including *DfFPS1*, *DfFPS2*, *DfGGPS1*, *DfGGPS2*, *DfGPS*, *DfHMGR1*, *DfTPS* and *DfIDI* (Figure 3, Appendix A).

### 2.5. Transcription Factors

In total, 221 significant differentially expressed transcription factors (TF) were obtained in CK compared with UV–B and belonged to 16 TF families. AP2/ERF(46), WRKY (20), bZIP (8), bHLH (20) and SPL (9) were mainly enriched (Figure 2A). In total, 159 TF were up-regulated and 62 TF were down-regulated, respectively. These transcription factors were involved in 7 cellular components, 8 biological processes and 10 molecular functions, mainly enriched in a transcription regulator complex (GO:0005667), regulation of transcription (GO:0006355) and protein dimerization activity (GO:0046983), respectively (Figure 2B). For KEGG enrichment, the TF were involved in a total of eight metabolic pathways, mainly enriched in the plant hormone signal transduction (ko04075) (Figure 2C). Among these GO terms and KEGGs, nine key genes were identified, including *DfSPL1, DfSPL2, DfDELLA, DfIAA9, DfGID1, DfJAZR1, DfCTR1, DfJAZ,* and *DfMYC2* (Figure 3, Appendix A).

### 2.6. Identification of Target Genes of Differentially Expressed miRNA in Tuber Expansion Compared with Initiation Stage

In total, 17 differentially expressed miRNAs (DEMs) were obtained in CK compared with UV–B, including 12 up-regulated DEMs and 7 down-regulated DEMs. There were eight known miRNAs belonging to five miRNA families (Appendix A). The most abundant DEMs with the highest number of expressions during UV–B were depicted by a heat map. The miR171 family (dfr–miRNA171a, dfr–miRNA171b, dfr-miRNA171c) and miRNA166 family (dfr–miRNA166a) were up-regulated during UV–B, miR156 family (dfr–miRNA156b, dfr–miRNA156c), miR408 family (dfr–miRNA408) and miR160 family (dfr–miRNA160a) were down-regulated compared to the control group. Research shows that miRNAs negatively regulate target mRNA through translation repression or mRNA degradation. Subsequently, we identified targets for DEMs using psRNA Target, 8 DEMs were putatively targeted to 115 DEGs (Appendix A). Auxin response factors (ARFs) were the target gene of dfr-miRNA160 (ARF17 and ARF18, targeted by miRNA160); and the transcription factor of SQUAMOSA promoter binding protein-like (SPL) gene was the target gene of dfr-miR156 (Figure 4). Furthermore, KEGG pathway enrichment analysis indicated that the target mRNAs were enriched in 34 pathways, mainly enriched in the plant hormone signal transduction (ko04075), phenylpropanoid biosynthesis (ko00940), flavonoid biosynthesis (ko00941), phenylalanine metabolism (ko00360), ubiquinone and other terpenoid-quinone biosynthesis (ko00130), starch and sucrose metabolism (ko00500) and glutathione metabolism (ko00480) (Appendix A). According to GO enrichment analysis, these transcription factors were involved in 11 cellular components, 33 biological processes, and 25 molecular functions, mainly enriched in oxidoreductase activity (GO:0016491), interspecies interaction between organisms (GO:0044419), viral process (GO:0016032), tetrapyrrole binding (GO:0046906), obsolete electron transport (GO:0006118), heme binding (GO:0020037), viral life cycle (GO:0019058), sequence-specific DNA binding (GO:0043565), cytoskeletal protein binding (GO:0008092), interaction with host (GO:0051701), electron carrier activity (GO:0009055) and photosynthesis (GO:0015979) (Appendix A). 

### 2.7. The dfr–miR156b–DfSPL3 Module Regulates the Expression of DfGGPS1

MiR156 was first discovered in *Arabidopsis thaliana*. It consists of about 20 nucleotides and is structurally conserved [35]. It is involved in regulating plant growth and development by targeting the SPL family [36]. To verify the target relationship between dfr–miR156b and *DfSPL3*, we performed transient co-transformation technology in tobacco (*Nicotiana benthamiana*). Compared with the Pro35S:: GUS leaves, Pro35S:: DfSPL3–GUS exhibited that the GUS phenotype was revealed by histochemical staining. The result showed that the *DfSPL3* was fused the *GUS* gene. Compared with Pro35S:: DfSPL3-GUS leaves, the GUS staining was markedly decreased in leaves co-transformed with the strain mixture Pro35S:: DfSPL3–GUS and Pro35S:: dfr-miR156b (Figure 5A). This was considered to be additional evidence that dfr–miR156b could target *DfSPL3*. To confirm the results of these histochemical observations, RNA was extracted after two days of co-expression in tobacco, and the expression of *DfSPL3* was analyzed by qRT–PCR. Compared with Pro35S:: DfSPL3-GUS leaves of DfSPL3 expression, the expression of DfSPL3 significantly declined in leaves co-transformed with the strain mixture Pro35S:: DfSPL3–GUS and Pro35S:: dfr–miR156b (Figure 5B). Taken together, these results showed that dfr–miR156b targets *DfSPL3*.

The GTAC motif has been identified as the core binding site of SPLs [37,38]. Within a 2000-bp fragment of the DfGGPS1 gene upstream of the translation start codon, there are six GTAC motifs. Compared with the Pro35S:: GUS leaves, ProDfGGPS1:: GUS exhibited that the GUS phenotype was revealed by histochemical staining. The result showed that the ProDfGGPS1 was activated *GUS* gene expression. Compared with ProDfGGPS1:: GUS leaves, the GUS staining was markedly increased in leaves co-transformed with the strain mixture ProDfGGPS1:: GUS and 35S:: DfSPL3 (Figure 5C). This was considered to be additional evidence that DfSPL3 could interact with and activate the DfGGPS1. In order to further confirm the specific activation effect of DfSPL3 on the *DfGGPS1* promoter, A Y1H verified that DfSPL3 could directly bind to the *DfGGPS1* promoter (Figure 5D). These results indicated that DfSPL3 could interact with and activate the *DfGGPS1*. 

### 2.8. qRT-PCR Analysis of DEGs and DEMs Data

To characterize the accuracy and reliability of the RNA-Seq and miRNA data, RT-qPCR was used to measure the expression of a number of DEGs and DEMs, including 10 mRNAs and 7 miRNAs, for which specific primers were designed. These included *DfGGPS1*, *DfGGPS2*, *DfDELLA*, *DfMYC2*, *DfSPL3*, *DfSPL6*, *DfSPL9*, *DfJAZ*, *DfFPS1*, *DfFPS2*, dfr–miRNA171a, dfr–miRNA171b, dfr–miRNA171c, dfr–miRNA156b, dfr–miRNA156c, dfr–miRNA160a and dfr–miRNA408. They were mainly enriched in the plant hormone signal transduction (ko04075), diterpenoid biosynthesis (ko00904), and terpenoid backbone biosynthesis (ko00900). The results showed that, compared with CK, the expression trends of *DfGGPS1*, *DfGGPS2*, *DfDELLA*, *DfMYC2*, *DfSPL3*, *DfSPL6*, *DfSPL9*, *DfJAZ*, *DfFPS1*, *DfFPS2*, dfr–miRNA171a, dfr–miRNA171b, dfr–miRNA171c, dfr–miRNA156b, dfr–miRNA156c, dfr–miRNA160a and dfr–miRNA408 in UV–B were consistent with their changes in RNA–Seq and miRNA data (Figure 6). These results indicated that the RNA–Seq and small RNA data were reliable.

## 3. Discussion

In recent years, with the destruction of the Earth’s ozone layer, the physiological and ecological studies of UV–B radiation on plants have received increasing attention from scholars [39]. UV–B radiation can cause the imbalance of ROS metabolism and produce a large amount of superoxide ions, which can intensify the peroxidation of membrane lipids and eventually cause damage to the membrane structure and normal physiological functions of plants [40]. The main enzymatic scavenging systems for ROS scavenging in plant cells are SOD, CAT and POD [41]. When plants are stressed, the activities of these antioxidant enzymes change accordingly and are involved in the regulation of ROS metabolism to protect plants from environmental stresses [42]. Studies have shown that UV–B has important effects on the expression of genes related to terpene biosynthesis in plants [43], with increased accumulation of terpene products being one of the more studied stress responses in the plant field. For example, carotenoids and chlorophyll, which are involved in plant photosynthesis, absorb and transmit light energy and are important for antioxidant activity in biofilms [6]. In this study, after exposure to UV–B stress, the activities of antioxidant enzymes (SOD, CAT and POD) were enhanced and the ROS content was reduced in fragrant woodfern of leaves. 

Terpenoids are the most structurally and quantitatively diverse group of plant secondary metabolites, and they play an important role in plants. Transcription factors regulate the production of terpenoids by regulating the expression of genes in the terpene metabolic pathway. Six families of transcription factors were found to be involved in terpenoid synthesis, including AP2/ERF, bHLH, MYB, NAC, WRKY, and bZIP [44]. In *Salvia miltiorrhiza SmMYB9b* enhances tanshinone content by promoting the expression of *SmDXS2*, *SmDXR*, *SmGGPPS,* and *SmKSL1* [45]. The *AnTAR1* transcription factor regulates artemisinin synthesis by participating in the initiation of artemisia leaf trichomes [46]. *CitERF71* enhances the synthesis of geraniol, an important volatile monoterpene in sweet orange (*Citrus sinensis*), by promoting the expression of *CitTPS16* [47]. In this study, AP2/ERF (46), WRKY (20), bZIP (8), bHLH (20), and SPL (9) were found to be differentially expressed in response to UV–B treatment. All data suggest that the expression of these transcription factors is associated with terpenoid formation. In the future, more experimental evidence will be needed to confirm the role of these candidate transcription factors. 

Plant hormones play an important role in the regulation of plant growth and development and secondary metabolism. Gibberellins (GAs) are diterpenoid phytohormones that regulate plant growth and a wide range of developmental processes throughout the life cycle. The DELLA protein family is a key component of GA signaling, with nuclear localization and acts as a negative regulator of plant growth [48]. The study found that *AtMYC2* can balance various hormone signals by interacting with *AtDELLA*, affecting the expression of hemiterpene synthase *AtTPS21/11*, and then regulating the production of sesquiterpene (E)-β-caryophyllene in *Arabidopsis thaliana* [48]. In the present study, DfMYC2 was up-regulated and *DfDELLA* was down-regulated under UV–B treatment. It is suggested that UV–B inhibits the expression of *DfDELLA* through the GA signaling pathway, which results in the release of active *DfMYC2*, thereby promoting the expression of downstream terpenoid genes. Recently, the DELLA protein RGA was reported to promote MYC2-dependent JA signaling by competitively binding to JAZ1 (jasmonate ZIM-domain) [49,50]. JAZ proteins in Arabidopsis have been shown to be transcriptional repressors in the JA signaling pathway, and they interact with and repress the function of the JA response gene MYC2 transcription factor [51]. When sufficient JA is synthesized in plants, it leads to the degradation of JAZ by the 26S protein hydrolase pair, allowing the release of active MYC2, which initiates downstream genes [52], regulating terpene synthesis or plant development. Our experimental results showed that *DfJAZ* gene expression was down-regulated under UV–B treatment. This suggests that UV–B treatment, which increases JA in vivo, leads to the degradation of JAZ by the 26S protein hydrolase pair, allowing the release of active *DfMYC2*, which initiates downstream genes that regulate terpene synthesis or plant development, for example. Overall, these results suggest that UV–B regulates terpenoid synthesis through hormone signaling pathways in a complex regulatory network.

miRNA-mediated gene regulation has been extensively studied in terpene synthesis via the transcriptional and post-transcriptional levels, which provides the basis for a better understanding of the UV–B regulated terpene synthesis network [53]. In the aromatic perennial herb patchouli (*Pogostemon cablin*), the patchouli alcohol synthase (*PatPTS*) gene is regulated by the squamosa promoter-binding protein-like (SPL) transcription factor targeted by miR156 [54]. In *Arabidopsis thaliana*, the miR156–SPL module regulates flowering b-stigmasterene formation by regulating the expression of the sesquiterpene synthase gene *AtTPS21* [54]. In the present study, dfr-miR156b was down-regulated in expression and its target gene SPL3, up-regulated in expression under UV–B treatment, indicating that UV–B promotes *DfTPS* gene expression through the dfr-miR156-SPL3 module. It has been reported that the auxin response factor 6 (*ARF6*) and *ARF8* trigger the expression of two jasmonic acid (JA)-inducible genes, *MYB21* and *MYB24*, by enhancing JA production, which in turn promotes petal, stamen, gynoecium, and nectary development and consequently affects sesquiterpene production [55,56]. In this study, *ARFs* were the target gene of dfr-miRNA160a. It was shown that UV–B is involved in mediating terpene production through dfr-miR160a affecting the *DfARF* gene.

In summary, many mRNAs and miRNAs work together to construct miRNA–mRNA networks for UV–B regulation of terpenoid metabolite synthesis (Figure 5E). Among these networks, the dfr-miR156-DfSPL3 module regulates the formation of terpenoid synthesis in UV–B treatment through modulating the expression of the *DfGGPS1*. These results suggested that miRNA–mRNA regulatory networks play a key role in terpene accumulation in fragrant woodfern leaves.

## 4. Materials and Methods

### 4.1. Plant Materials and Treatment

Fragrant woodfern (*Dryopteris fragrans*) seedlings were propagated from the laboratory of plant resources and molecular biology of Northeast Agricultural University. They were grown in containers containing a 1:1 mixture of vermiculite and grass charcoal soil at 25 ± 2 °C with a 16 h-light and 8 h-dark photoperiod. For all the physiological and molecular experiments, there were two groups of acclimatized 12-week-old seeds. First group served as the control and plants from the second group were exposed to UV–B (2.8 W m^−2^) radiation, which was provided daily for 1 h at noon for 6 days. UV–B was artificially provided by G15T8E UV–B lamps (Sankyo Denki co., Ltd., Sapporo City, Japan). The lamps were held in a movable frame over the plants. The lamps were wrapped with 0.125 mm-thick cellulose diacetate film (Shanghai Plastic Company, Shanghai, China) to filter out the UV–C (<280 nm) radiation. Control plants were also kept under the same lamp but covered with polyester film that excludes radiations below 320 nm. The UV–B irradiance at the top of the plantlet was measured with an ultraviolet intensity meter (UV P. Inc., San Gabriel, CA, USA).

### 4.2. Enzyme-Labeled Instrument Analysis of Photosynthetic Pigments

Photosynthetic pigments including chlorophyll a (Chl a), chlorophyll b (Chl b) and total carotenoids were measured using previously published techniques [57]. Briefly, 0.2 g of fresh fragrant woodfern leaf samples were minced, homogenized in a 3 mL solution containing 95% acetone and a limited quantity of CaCl_2_ and quartz sand, and were then spun for 30 min at 5000× *g* at 4 °C. Supernatants were then collected, and levels of Chl a, Chl b, and carotenoids therein were measured at absorbance wavelengths of 665, 649, and 470 nm, respectively, using an enzyme-labeled instrument (Thermo Scientific, Waltham, MA, USA). Levels of these three pigments were then quantified as follows: chlorophyll a = 13.95·A665–6.88·A649, chlorophyll b = 24.96·A649–7.32·A665 and total carotenoids = (1000·A470–2.05·Chl a–114.8·Chl b)/245.

### 4.3. Measurement of Cell Death

Cell death indicated as a loss of plasma membrane integrity was evaluated spectrophotometrically as Evans blue uptake [22]. Leaf tissues (0.1 g fresh weight) were incubated in Evans blue solution (0.25% (*w*/*v*) Evans blue (Sigma, St. Louis, MI, USA) in water) for 1 h at room temperature. After washing with distilled water for 15 min, the trapped Evans blue was released from the leaves by homogenizing leaf tissue with 1 mL of 1% (*w*/*v*) aqueous SDS. The homogenate was centrifuged at 14,000× *g* for 15 min. The optical density of the supernatant was determined at 600 nm by using Shimadzu UV-2600 spectrophotometer.

### 4.4. Assay on ROS and Peroxidase Activity in Fragrant Woodfern Plants 

The ROS and peroxidase activity in fragrant woodfern seedlings were measured after UV–B stress. The H_2_O_2_ content (H_2_O_2_–1–Y), O_2_^−^ content (SA–1–G), MDA content (MDA–1–Y), SOD activity (SOD–1–Y), CAT activity (CAT–1–W), and POD activity (POD–1–Y) were measured using a kit (Comin Biotechnology, Suzhou, China) and read optical density at 415 nm, 530 nm, (532 and 600 nm), 450 nm, 405 nm and 470 nm, respectively. The content of H_2_O_2_ was determined by the formation of a yellow complex ([TiO(H_2_O_2_)]_2_) by H_2_O_2_ and TiSO_4_(TiO_2_). After the complex can be dissolved by H_2_SO_4_, which has a characteristic absorption peak at 415 nm. The content of O_2_^−^ determined the formation of NO_2_^−^ by O_2_^−^ reaction with hydroxylamine. NO_2_^−^ continues to react with p-aminobenzenesulfonic acid and α-naphthylamine to form a red compound (p-phenylsulfonic acid-azo-α-naphthylamine), which has a characteristic absorption peak at 530 nm and the content of O_2_^−^ chemical calculation is directly carried out according to the reaction formula. MDA can be condensed with thiobarbital acid (TBA) under higher temperature (95 °C) and acidic conditions to form a red MDA-TBA complex, which has a characteristic absorption peak at 532 nm at the same time, the absorbance at 600 nm is measured, and the difference between the absorbance at 532 nm and 600 nm is used to calculate the MDA content. The SOD activity was determined by the reaction system of xanthine and xanthine oxidase to produce s O_2_^−^. O_2_^−^ reduces nitro blue tetrazolium to produce blue methionine (NBT), which has a characteristic absorption peak at 560 nm. SOD can remove O_2_^−^, thus inhibiting the formation of methionine. The CAT activity was determined by the characteristic absorption peak of H_2_O_2_ at 240 nm. CAT can decompose H_2_O_2_, so that the absorbance of the reaction solution at 240 nm decreases with the reaction time. The CAT activity was calculated based on the change rate of the absorbance. The POD activity determined the formation of brown complex 4-o-methylphenol as POD-catalyzed H_2_O_2_ oxidizes guaiacol, which has a characteristic absorption peak at 470 nm.

### 4.5. Determination of the Secondary Metabolite Content of Dryopteris Serrata by GC–MS

#### 4.5.1. Sample Processing before Sample Loading in GC–MS Experiment

Put 0.1 g of fragrant woodfern leaves and 1 mL of ethyl acetate standard solution into a 1.5 mL EP tube, which contained 8.64 µg.mL^−1^ nonyl acetate as the internal standard. Put the EP tube into an ultracentrifugal mill (700 r min^−1^, 5 min). Put the ground EP tube into a small ultrasonic cleaner for ultrasonic treatment (60 kHz, 40 °C, 30 min). Put the sonicated EP tube into a low-temperature centrifuge (5000× *g*, 5 min). Aspirate the supernatant and filter it through a 0.22 µm microporous membrane for GC-MS experimental detection.

#### 4.5.2. GC–MS Instrument Experimental Conditions

Terpenoid detection was analyzed using an Agilent 7890A gas chromatograph coupled to an Agilent 5975C Network Mass Selective Detector (MS, insert XL MSD with triple-axis detector). Chromatographic column: HP-5MS column (30 m × 0.25 mm × 0.25 μm; J&W Scientific, Folsom, CA, USA); Carrier gas: He (purity ≥ 99.999%); Injection port temperature: 280 °C; Injection method: Splitless injection; Injection volume: 1 µL. Program temperature rise: the initial temperature is 60 °C for 2 min, the temperature is increased at 20 °C/min to 220 °C for 1 min, the temperature is increased at 5 °C.min^−1^ to 250 °C for 1 min, and finally the temperature is increased at 20 °C.min^−1^ to 290 °C for 7.5 min.

Ion source: EI; ion source temperature: 230 °C; ion energy mode: use tuning setting; ion energy (eV): 70; detector setting: use gain factor; solvent delay: 5 min; mass scan range: 30~500.

### 4.6. RNA Extraction, Library Construction and Sequencing

Fragrant woodfern seedlings were used for total RNA extraction using MiniBESTreagent (TaKaRa, Dalian, China). The extracted RNA was treated with DNase I (TaKaRa, Dalian, China) to remove the contaminated DNA and detected by 1.0% agarose gel electrophoresis. RNA integrity was assessed using the RNA Nano 6000 Assay Kit of the Bioanalyzer 2100 system (Agilent Technologies, Santa Clara, CA, USA).

For RNA-seq, 3 μg of total RNA from each sample was used for library preparation using a NEBNext^®^ Ultra™ directional RNA library prep kits (NEB, code no. E7420S). RNA was fragmented into small pieces and then first-strand cDNA was synthesized with SuperScript II reverse transcription (Invitrogen, Carlsbad, CA, USA). After purification, the second-strand cDNA library was synthesized, following several rounds of PCR amplification. PCR products were purified (AMPure XP system) and library quality was assessed on the Qubit2.0 Fluorometer, Agilent Bioanalyzer 2100 system and qRT-PCR. Quantified on a 150 bp paired-end run by Agilent2200 and sequenced by Novogene (Novogene, Beijing, China) on an Illumina Hiseq2500 platform.

For small RNA sequencing, small RNA libraries using an NEB Multiplex Small RNA Library Preparation Kit (NEB, Ipswich, MA, USA). Briefly, 5 μg of total RNA was ligated to a 5′ RNA adaptor and 3′ RNA, small interfering RNA, and piwi-interacting RNA, and first-strand cDNA were then synthesized using M-MuLV reverse transcriptase (RNase H-) as a catalyst. PCR amplification was performed with specific primers and LongAmp Taq 2× Master Mix (Illumina, San Diego, CA, USA). The RNAs were reverse transcribed to cDNAs, following PCR amplification. Subsequently, the libraries were purified and sequenced by Novogene (Novogene, Beijing, China) on an Illumina Hiseq2500 platform with 125 bp paired-end and 50 bp single-end, respectively. Three biological replicates were performed for each sample.

### 4.7. Identification and Functional Annotation of Differentially Expressed Genes and miRNAs

The expression levels of mRNAs were measured as fragments per kilobase of exon per million fragments mapped (FPKM), and genes with expression levels > 5 FPKM were retained for statistical analysis. miRNA read counts were normalized to reads per million transcripts (TRM). Differentially expressed genes (DEGs) and miRNAs (DEMs) were identified by DEGseq after significance [58]. *p*-values and false discovery rate adjusted *p*-value (FDR) analyses were performed at absolute values of log2FC ≥ 1, *p* < 0.05, FDR < 0.05 [59]. Functional databases NT, NR, Eukaryotic Ortholog Groups (KOG), and SwissProt (http://www.gpmaw.com/html/swiss-prot.html (accessed on 1 May 2021)) were used to annotate gene function, while Blastn, Blastx, Diamond, Blast2GO [60], and InterProScan5 were used to align genes. To detect domains in the translated protein sequences, we used Pfam (http://pfam.xfam.org/ (accessed on 1 May 2021)). All DEGs were subjected to gene ontology (GO, http://geneontology.org/ (accessed on 1 May 2021)) and Kyoto Encyclopedia of Genes and Genomes (KEGG, https://www.kegg.jp/ (accessed on 1 May 2021)) pathway analysis. To identify significant GO and KEGG pathway categories, Fisher’s exact tests were applied under absolute values of *p* < 0.05 and FDR < 0.05 [61]. Each DEG was predicted by aligning the gene sequences against the Plant Transcriptional Factor Database (PlantTFDB, http://planttfdb.gao-lab.org/ (accessed on 1 May 2021)). The DEGs were classified according to their TF families. A miRNA-target gene regulatory network was constructed using Cytoscape_v3.2.1 program. Clustering analysis was performed using MeV 4.9.0 with the Pearson correlation.

### 4.8. Validation of the DEGs and DEMs Data Using qRT-PCR

The total RNA was reverse transcribed into first-strand cDNA using HiScript^®^ III 1st Strand cDNA Synthesis Kit (Vazyme, Nanjing, China). RNA removal DNA contamination system was 10 μL, including 8 μL RNA and 2 μL 5 × gDNA wiper Mix. Its conditions were as follows: 42 °C for 2 min. The reverse transcribed into first-strand cDNA system was 20 μL including 10μL of RNA removal DNA, 2 μL 2 × RT Mix, 2 μL HiScript III Enzyme Mix, 1 μL Oligo (dT)_20_VN, and 5 μL RNase-free ddH_2_O. Its conditions were as follows: 25 °C for 5 min, 37 °C for 45 min, and 85 °C for 5 s. 

The total RNA was reverse transcribed into miRNA First-Strand using Mir-X™ miRNA First-Strand Synthesis Kit (TaKaRa, Dalian, China). miRNA First-Strand System was 10 μL, including 5 μL 2× mRQ Buffer, 3.75 μL RNA sample (0.25–8 μg), and 1.25 μL mRQ Enzyme. Its conditions were as follows: 37 °C for 1 h, and 85 °C for 5 min. 

qRT-PCR was used ChamQ Universal SYBR qPCR Master Mix Kit (Vazyme, Nanjing, China); qRT-PCR system was 20 μL, including 10 μL 2 × ChamQ Universal SYBR qPCR Master Mix, 0.4 μL forward primer, 0.4 μL reverse primer, 30 ng of cDNA per sample. Its conditions were as follows: 95 °C for 30 s, followed by 40 cycles of 95 °C for 10 s, and 60 °C for 30 s. The reference gene selected for normalization in this experiment was *Df18SrRNA*. The relative expression levels of the genes were calculated by the 2^−ΔΔCt^ method, DEGs and DEMs of the primer sequence information were shown in Appendix A.

### 4.9. One-Hybrid Screening

Y1H assay was carried out using the Matchmaker One-Hybrid System of Clontech, as described in the manufacturer’s protocol. *DfGGPPS1* transcriptional promoter was inserted into pHIS2 vector to generate the pHIS2–*ProGGPPS1* recombinant construct. Full-length *DfSPL3*-encoding sequence was inserted into pGADT7 to construct the pGADT7–*DfSPL3* vector. The pGADT7–*DfSPL3* and pHIS2–*ProGGPPS1* were transformed into Y187, respectively. Then, it was screened on selective SD/-Leu/-Trp (DDO) medium uracil. Colony PCR analysis was used to confirm that the plasmids had integrated correctly into the genome of Y187. After determining the minimal inhibitory concentration of 3-AT (3-Amino-1, 2, 4-triazole) for the bait strains, the pHIS2-*ProGGPPS1* vectors were transformed into the bait strain and screened on an SD/-His/-Leu/-Trp (TDO) plate. The primers used are listed in Appendix A. All transformations and screenings were performed three times. 

### 4.10. Agrobacterium-Mediated Transient Transformation in Tobacco (Nicotiana Benthamiana)

Full-length dfr–miR156b-encoding sequence was inserted into pCAMBIA2301 vector to generate the Pro35S:: dfr–miR156b recombinant construct. Full-length *DfSPL3*-encoding sequence was inserted into pBI121 vector to generate the Pro35S:: *DfSPL3*–GUS recombinant construct, which was fused with DNA sequences β-Glucuronidase (GUS). Pro35S:: dfr–miR156b, Pro35S:: *DfSPL3*–GUS, and Pro35S:: GUS constructs were introduced into Agrobacterium tumefaciens strain LBA4404. Then, tobacco leaves (*Nicotiana benthamiana*) were injected with Agrobacterium-harboring constructs. After 2 d of β-Glucuronidase (GUS), staining and GUS quantitative detection were conducted 2 d after infiltration. Full-length *DfSPL3*-encoding sequence was inserted into pCAMBIA2301 vector to generate the Pro35S:: *DfSPL3* recombinant construct. *DfSPL3* promoter sequence was inserted into pBI121 vector to generate the *ProDfGGPS1*:: GUS recombinant construct, which was fused with DNA sequences β-Glucuronidase (GUS). *ProDfGGPS1*:: GUS and Pro35S:: *DfSPL3* constructs were introduced into Agrobacterium tumefacien strains LBA4404. Then, tobacco leaves (*Nicotiana benthamiana*) were injected with Agrobacterium-harboring constructs. After 2 days of β-Glucuronidase (GUS), staining detection was conducted 2 days after infiltration. The primers used are listed in Appendix A. Each experiment was performed with at least three independent replicates. 

### 4.11. Statistics Analysis

Excel was used for data analysis, and GraphPad Prism 8.0 was used to prepared Figures. The data were performed independently at least three times and expressed as means ± SEM. Statistical analysis was assessed by one-way ANOVA. Asterisks indicate significant differences revealed by one-way ANOVA at *p* < 0.05 (*) and *p* < 0.01 (**), respectively.

## 5. Conclusions

mRNA and miRNA-seq identified 17 DEMs and 4533 DEGs. mRNA–miRNA analysis identified miRNA target gene pairs consisting of 8 DEGs and 15 miRNAs. The target genes were mainly enriched in diterpenoid biosynthesis, terpenoid backbone biosynthesis, plant hormone signal transduction, MVA pathway, and MEP pathway, among which dfr-miR156b and dfr-miR160a These pathways are regulated through the targeting of *DfSPL* and *DfARF*, respectively. The mRNA and miRNA datasets identified a subset of candidate genes. We found that mRNA–miRNAs were involved in UV–B regulation of terpenoid synthesis in fragrant woodfern. However, we proposed a hypothetical model that UV–B regulates the terpenoid synthesis of the genetic regulatory network in fragrant woodfern. It provides the theoretical basis that UV–B regulates the terpenoid synthesis of the molecular regulatory mechanism in fragrant woodfern.

## Figures and Tables

**Figure 1 ijms-23-05708-f001:**
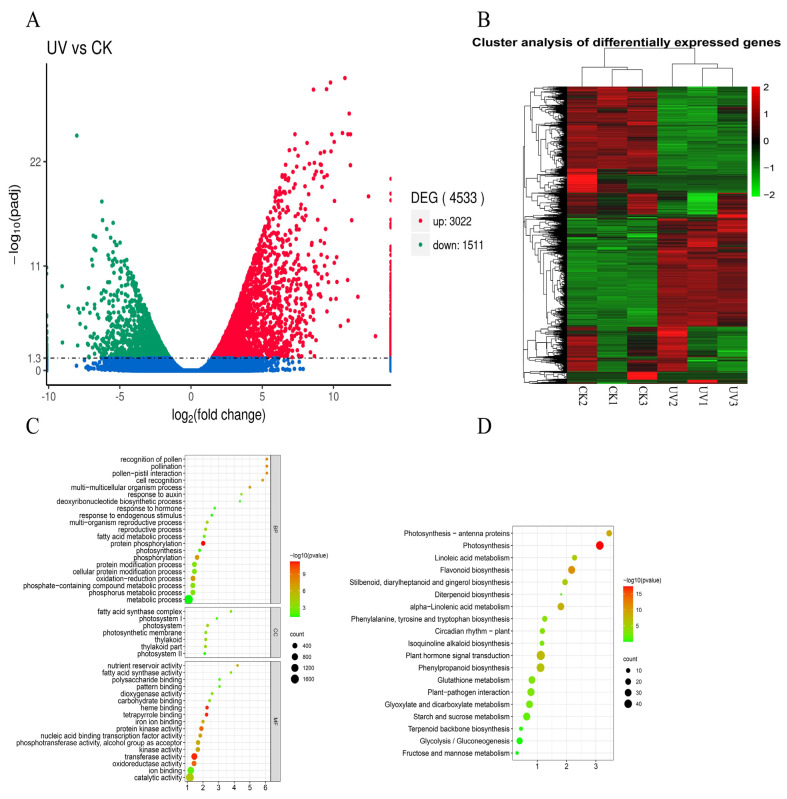
Differentially expressed genes were obtained under UV–B treatment in fragrant woodfern leaves. (**A**) Volcano map representing the levels of DEGs; (**B**) Heat map representing the levels of DEGs; (**C**) GO functional classification of DEGs; (**D**) KEGG functional classification of DEGs.

**Figure 2 ijms-23-05708-f002:**
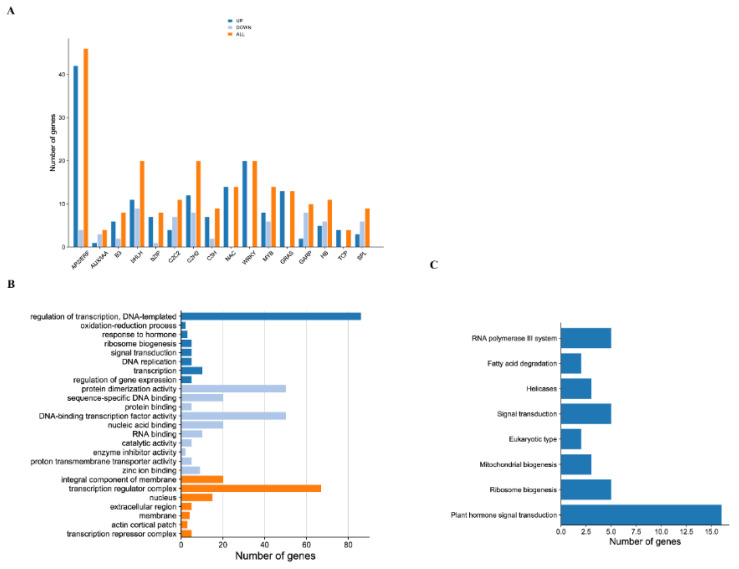
Transcription factors were obtained in Fragrant woodfern leaves. (**A**) The number of up-regulated and down-regulated transcription factors during UV–B; (**B**) GO functional classification of transcription factors; (**C**) KEGG functional classification of transcription factors.

**Figure 3 ijms-23-05708-f003:**
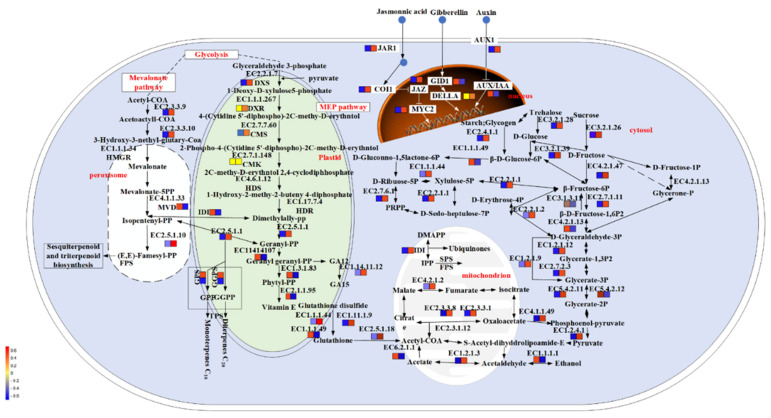
Related pathways of UV–B regulation of terpenoid synthesis in fragrant woodfern leaves. Identified DEGs and transcription factors detected in these pathways. The icons beside each gene name show the change in the gene differential expression level.

**Figure 4 ijms-23-05708-f004:**
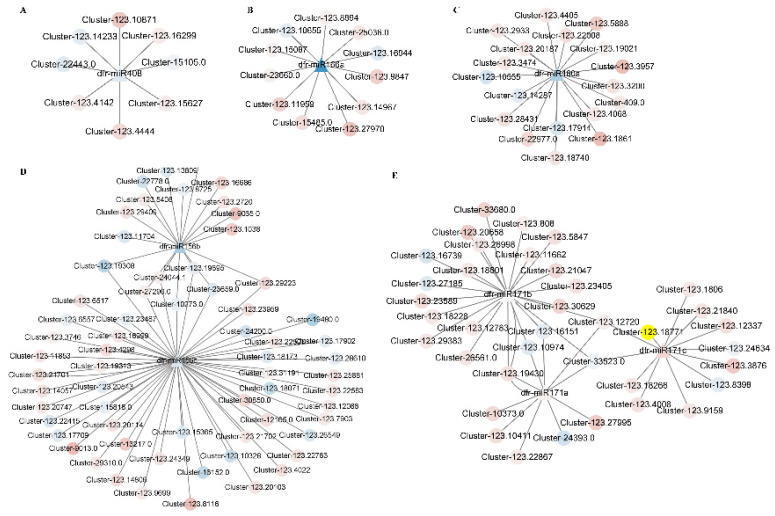
Regulatory network from the integrated analysis of miRNA–mRNA data. (**A**) dfr–miRNA408–mRNAs of regulatory network. (**B**) dfr–miRNA166a–mRNAs of regulatory network. (**C**) dfr–miRNA160a–mRNAs of regulatory network; (**D**) dfr–miRNA156b/c–mRNAs of regulatory network. (**E**) dfr–miRNA171a/b/c-mRNAs of regulatory network; Co-expressed miRNA–mRNA interactions visualized as a network using Cytoscape. The circle represents mRNA, the triangle represents miRNA, red represents up-regulation, and blue represents down-regulation in the network.

**Figure 5 ijms-23-05708-f005:**
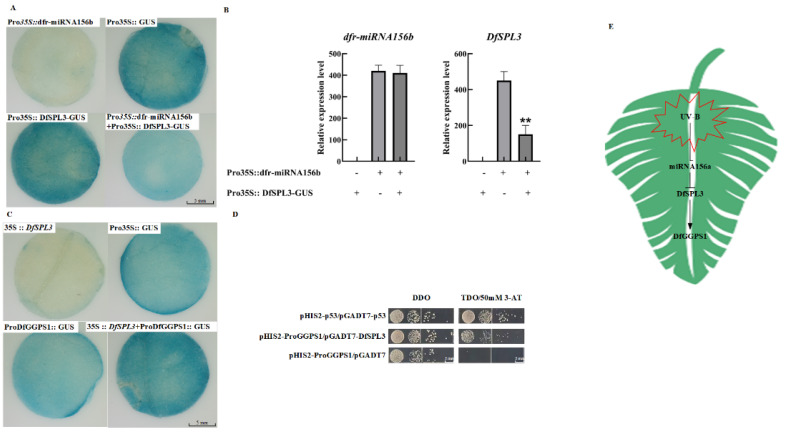
(**A**) β-Glucuronidase (GUS) phenotype observed by histochemical staining analysis of dfr-miR156b targets *DfSPL3*. (**B**) Co-expression of the constructs containing Pro35S:: DfSPL3–GUS and Pro35S:: dfr-miR156b in tobacco leaves. Expression levels determined by qPCR were normalized to the expression levels of tobacco. (**C**) β-Glucuronidase (GUS) phenotype observed by histochemical staining analysis of *DfSPL3* binding to the *DfGGPS1* promoter. (**D**) Yeast one-hybrid analysis of DfSPL3 binding to the DfGGPS1 promoter. Interaction was determined on SD/-His/-Leu/-Trp (TDO) medium lacking leucine in the presence of 3AT. pGADT7-p53 and pHIS2-p53 were used as positive controls. pHIS2-empty and pHIS2–ProDfGGPS1 were used as negative controls. (**E**) The simplified schematic networks of miRNA–mRNA networks for UV–B regulation of terpenoid metabolite synthesis. Asterisks indicate significant differences revealed by one-way ANOVA at *p* < 0.01 (**), respectively.

**Figure 6 ijms-23-05708-f006:**
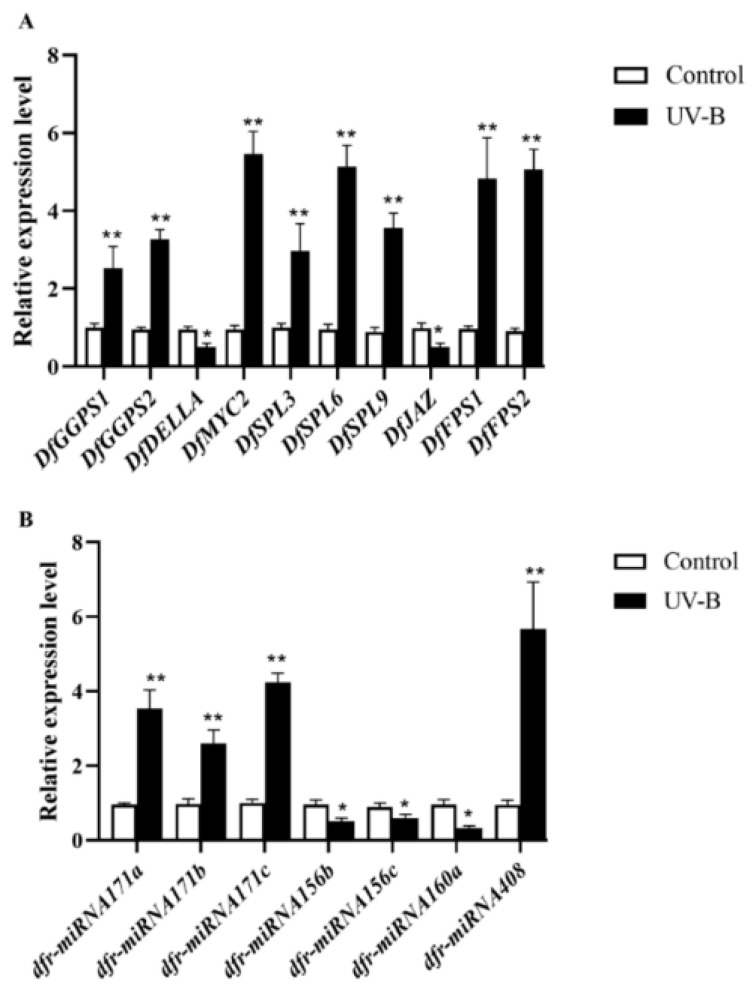
qRT–PCR analysis of DEG (**A**) and DEM (**B**) data. Asterisks indicate significant differences revealed by one-way ANOVA at *p* < 0.05 (*) and *p* < 0.01 (**), respectively.

**Table 1 ijms-23-05708-t001:** Statistical analysis of clean reads for mRNA in fragrant woodfern leaves.

Sample	Total	CK1	CK2	CK3	UV1	UV2	UV3
raw_reads	-	24,839,653	20,892,938	22,535,960	20,967,576	22,164,559	23,328,773
clean_reads	-	23,961,150	20,220,033	21,734,750	20,252,088	21,327,355	22,519,655
clean_bases	-	7.19G	6.07G	6.52G	6.08G	6.40G	6.76G
error_rate (%)	-	0.02	0.02	0.02	0.03	0.03	0.03
Q20 (%)	-	98.12	98.11	98.12	97.88	97.85	97.66
Q30 (%)	-	94.42	94.36	94.39	94.01	93.81	93.44
GC_pct (%)	-	46.92	46.03	45.9	48.37	47.32	47.3
N50	2035	-	-	-	-	-	-
N90	526	-	-	-	-	-	-
Number of transcripts	141,963	-	-	-	-	-	-
Number of Unigenes	69,493	-	-	-	-	-	-

**Table 2 ijms-23-05708-t002:** Statistical analysis of clean reads for small RNA sequencing in fragrant woodfern leaves.

Sample	Total Reads	Clean Reads	Mapped sRNA	Known miRNA	Novel miRNA	Total miRNA
CK1	15,490,143 (100.00%)	14,585,008 (94.16%)	6,307,041 (58.03%)	4	72	76
CK2	12,790,994 (100.00%)	12,417,161 (97.08%)	4,561,120 (51.72%)	6	71	77
CK3	15,716,150 (100.00%)	15,055,676 (95.80%)	6,417,540 (67.93%)	5	72	77
UV1	17,010,191 (100.00%)	16,386,263 (96.33%)	6,917,716 (61.08%)	6	77	83
UV2	13,075,551 (100.00%)	12,297,545 (94.05%)	4,580,695 (57.97%)	7	79	86
UV3	11,728,662 (100.00%)	11,077,797 (94.45%)	4,023,081 (54.21%)	6	78	84
Sum	85,811,691	81,819,450				

## Data Availability

Not applicable.

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
