# Peer review of "Integrated mRNA and miRNA Transcriptome Analysis Suggests a Regulatory Network for UV–B-Controlled Terpenoid Synthesis in Fragrant Woodfern (Dryopteris fragrans)"

_ijms, 2022, doi:10.3390/ijms23105708_

Round 1

Reviewer 1 Report

Some improvements have been made in this resubmission, but there are still many problems. For example, the "Figure 4" is missing. Units and numbers should have a space in the text, except "℃" and "%". Authors need to double-check the corrections.

As for the experimental results, we still do not see the contents of GC-MS (for secondary metabolites) in the "Supplementary Table", and the author still does not have any data to show the experimental results. At least it is better to attach the chromatogram figure (GC-MS).

My suggestion is major revision.

Author Response

Response to Reviewer 1 Comments

Point 1: Some improvements have been made in this resubmission, but there are still many problems. For example, the "Figure 4" is missing.

Response 1:

Thank you very much for your question and help us to improve the quality of our manuscript.

manuscript has been modified.

Figure 4. Regulatory network from the integrated analysis of miRNA-mRNA data. A, dfr-miRNA408-mRNAs of regulatory network; B, dfr-miRNA166a-mRNAs of regulatory network; C, dfr-miRNA160a-mRNAs of regulatory network; D, dfr-miRNA156b/c-mRNAs of regulatory network; E, dfr-miRNA171a/b/c-mRNAs of regulatory network; Co-expressed miRNA-mRNA interactions visualized as a network using Cytoscape. the circle represents mRNA, the Triangle represents miRNA, red represents up-regulation and blue represents down-regulation in network.

Point 2: Units and numbers should have a space in the text, except "℃" and "%". Authors need to double-check the corrections.

Response 2:

Thank you very much for your question and help us to improve the quality of our manuscript.

manuscript has been modified.

Point 3: As for the experimental results, we still do not see the contents of GC-MS (for secondary metabolites) in the "Supplementary Table", and the author still does not have any data to show the experimental results. At least it is better to attach the chromatogram figure (GC-MS).

Response 3:

Thank you very much for your question and help us to improve the quality of our manuscript.

Supplementary Table and Figure has been modified.

Supplementary Table 1 GC-MS detection of contents of terpenoid in Fragrant Woodfern

min

CAS No

description

Molecular Formula

0

2

4

6

CK

UV-B

CK

UV-B

CK

UV-B

CK

UV-B

1

5.341

31983-22-9

α-Muurolene

C15H24

2.85±0.05

5.37±1.25

1.68±1.01

3.29±0.2

0.00

1.96±0.02

0.41±0.02

3.08±1.02

2

6.848

116-04-1

β-HUMULENE

C15H24

2.07±0.07

1.46±0.9

1.18±1.02

1.35±0.02

0.93±0.02

1.85±0.08

1.33±0.12

1.45±0.07

3

7.988

544-76-3

hexadecane

C16H34

2.12±0.1

1.90±0.7

1.88±0.9

2.20±0.1

1.52±0.02

3.13±0.9

1.98±0.05

3.14±1.02

4

8.209

143-13-5

nonyl acetate

C11H22O2

8.64±0.9

8.64±1.25

8.64±1.08

8.64±1.02

8.64±1.02

8.64±1.02

8.64±1.02

8.64±1.07

5

8.274

540-97-6

Cyclohexasiloxane,dodecamethyl

C12H36O6Si6

2.15±1.01

1.73±0.7

1.09±0.05

1.66±0.01

1.03±0.01

1.68±0.01

1.28±0.04

1.31±0.07

6

8.344

3891-98-3

2,6,10-trimethyldodecane

C15H32

0.00

0.00

0.89±0.05

1.01±0.07

0.85±0.01

1.35±0.02

0.95±0.07

1.33±0.04

7

8.49

6831-16-9

(-)-ARISTOLENE

C15H24

2.92±0.9

1.70±0.9

1.87±1.01

2.00±0.2

1.34±0.02

2.55±0.01

1.71±0.07

2.34±0.5

8

8.663

17334-55-3

(+)-CALARENE

C15H24

1.05±0.9

0.00±

1.10±0.02

1.87±0.05

0.57±0.02

1.30±0.03

1.01±0.01

1.54±0.04

9

8.776

1139-17-9

(-)-ISOLONGIFOLOL

C15H26

1.84±0.7

1.31±0.04

2.36±0.9

1.41±0.05

1.26±0.01

1.43±0.05

2.16±0.07

3.29±0.07

10

9.225

26620-71-3

Naphthalene,1,2,3,4,6,7,8,8a-octahydro-1,8a-dimethyl-7-(1-methylethenyl)-, (1R,7R,8aS)-

C15H24

2.35±1.02

1.80±0.07

2.51±0.09

2.27±0.04

1.64±0.01

3.72±0.9

2.49±0.09

4.11±1.02

11

11.883

19078-37-6

1,4,4a,5,6,7,8,8a-Octahydro-2,5,5,8a-tetramethyl-1-naphthalenemethanol

C15H26

13.08±2.07

8.86±1.02

7.56±1.02

12.57±1.02

6.68±1.02

21.17±2.05

12.64±1.02

17.58±2.05

12

13.39

72345-17-6

1H-Cyclopropa[a]naphthalene, 1a,2,3,3a,4,5,6,7b-octahydro-1,1,3a,7-tetramethyl-, (1aS,3aR,7bR)-

C15H24

9.25±1.08

32.90±2.05

17.46±2.5

32.29±5.04

23.83±2.05

34.75±3.05

22.28±2.07

23.12±7.07

13

17.059

192724-27-9

(-)-drimenol

C15H26O

0.00

1.78±1.02

0.00

1.98±0.2

1.78±0.05

4.16±0.9

2.84±0.07

4.26±0.9

14

18.959

523-47-7

β-Cadinene

C15H24

3.04±1.01

3.00±0.9

3.72±1.02

4.00±1.02

3.57±0.9

5.65±0.9

6.81±1.02

5.23±0.9

15

19.591

112-95-8

N-EICOSANE

C20H42

0.00

1.69±0.07

1.91±0.01

2.08±0.02

1.64±0.02

3.98±0.8

2.30±0.07

3.95±0.9

SUM

51.35±5.7

72.13± 2.87 **

53.839±8.02

78.622±6.09

**

55.29±5.02 **

97.32±12.25

**

68.83 ±9.02

**

84.37±10.2**

Supplementary Figure 3. Effect of different UV-B radiation time on the contents of terpenoid in Fragrant Woodfern seedlings. A, Control group of 0 day on the contents of terpenoid in Fragrant Woodfern seedlings; B, UV-B radiation of 0 day on the contents of terpenoid in Fragrant Woodfern seedlings; C, Control group of 2 day on the contents of terpenoid in Fragrant Woodfern seedlings; D, UV-B radiation of 2 day on the contents of terpenoid in Fragrant Woodfern seedlings; E, Control group of 4 day on the contents of terpenoid in Fragrant Woodfern seedlings; F, UV-B radiation of 4 day on the contents of terpenoid in Fragrant Woodfern seedlings; G, Control group of 6 day on the contents of terpenoid in Fragrant Woodfern seedlings; H, UV-B radiation of 6 day on the contents of terpenoid in Fragrant Woodfern seedlings;

Reviewer 2 Report

This work provided useful information that mRNA-miRNAs were involved in UV-B regulates terpenoids synthesis in Fragrant Woodfern. There are two issues need to be addressed:

  1. In Table 2, “Tota reads” should be “Total reads”.
  2. The authors please carefully check the abbreviations/full names in the reference part, e.g. ref. 57. “…Journal of Plant Physiology 1994, 5, 28-29.”, and correct them.

Author Response

Response to Reviewer 2 Comments

Point 1: In Table 2, “Tota reads” should be “Total reads”.

Response 1:

Thank you very much for your question and help us to improve the quality of our manuscript.

Table 2 has been modified.

Table 2. Statistical analysis of clean reads for small RNA sequencing in Fragrant Woodfern leaves

Sample

Total reads

Clean reads

Mapped sRNA

Known miRNA

Novel miRNA

Total miRNA

CK1

15490143 (100.00%)

14585008 (94.16%)

6307041 (58.03%)

4

72

76

CK2

12790994 (100.00%)

12417161 (97.08%)

4561120 (51.72%)

6

71

77

CK3

15716150 (100.00%)

15055676 (95.80%)

6417540 (67.93%)

5

72

77

UV1

17010191 (100.00%)

16386263 (96.33%)

6917716 (61.08%)

6

77

83

UV2

13075551 (100.00%)

12297545 (94.05%)

4580695 (57.97%)

7

79

86

UV3

11728662 (100.00%)

11077797 (94.45%)

4023081 (54.21%)

6

78

84

Sum

85811691

81819450

Point 2:The authors please carefully check the abbreviations/full names in the reference part, e.g. ref. 57. “…Journal of Plant Physiology 1994, 5, 28-29.”, and correct them.

Response 2:

Thank you very much for your question and help us to improve the quality of our manuscript.

Ref has been modified.

  1. Wellburn, A.R. The Spectral Determination of Chlorophylls a and b, as well as Total Carotenoids, Using Various Solvents with Spectrophotometers of Different Resolution. J PLANT PHYSIOL 1994, 5, 28-29.

Round 2

Reviewer 1 Report

If the resolution of "Supplementary Figure 3" can be clearer, I think it should be acceptable by International Journal of Molecular Sciences.

This manuscript is a resubmission of an earlier submission. The following is a list of the peer review reports and author responses from that submission.

Round 1

Reviewer 1 Report

This article “Integrated mRNA and miRNA transcriptome analysis reveals a regulatory network for UV-B regulated the terpenoids synthesis in Fragrant Woodfern (Dryopteris fragrans)”. The comments for this manuscript are as follows:

  1. There are many errors in the section of "Refererences". The writing of references should be consistent, and each word should not be capitalized. Please according to the "Instructions for Authors" to rewrite the references.
  2. This manualscript seems to have great findings and results, but in fact the reviewer does not think so. In the section of “Materials and Methods”, the first the method for determination of chlorophyll was not found in reference. (Alan et al., 1994). Secondly, the brand and model of GC-MS and chromatography column are not recorded in detail, which will cause great trouble for readers. Although the experimental results seem to be rich, they are very poor in terms of evidence and discussion. In fact, if these mRNAs are all involved in their UV-B response, why the authors didn't find the proteins to prove it? It seems that these experimental results are only preliminary predictions, not actual results.
  3. In this manualscript, the authors found that mRNA-miRNAs are involved in UV-B regulation of the synthesis of terpenoids in Fragrant Woodfern, but what is the content of terpenoids? What experimental results can the authors rely on to prove the results of Figure 3? At the same time, the resolution of Figure 3 is too poor and should be reproduced.
  4. Are the results in Figure 5 evidence of protein? Only mRNA does not necessarily produce proteins. It is right that the synthetic pathway behind it is different. I don’t know why the authors have no considered this point.
  5. In the discussion, it due to the insufficient experimental results, the reviewer believes that it is not detailed enough and is of limited help to readers. In fact, the mechanisms by which plants synthesize the secondary metabolites is quite complicated. The authors mentioned the cell death of fern leaves in the results. How can readers believe that the latter data are meaningful?

My suggestion is major revision.

Author Response

Response to Reviewer 1 Comments

Point 1: There are many errors in the section of "Refererences". The writing of references should be consistent, and each word should not be capitalized. Please according to the "Instructions for Authors" to rewrite the references.

Response 1:

Thank you very much for your question and help us to improve the quality of our manuscript.

References has been modified.

Point 2:This manualscript seems to have great findings and results, but in fact the reviewer does not think so. In the section of “Materials and Methods”, the first the method for determination of chlorophyll was not found in reference. (Alan et al., 1994). Secondly, the brand and model of GC-MS and chromatography column are not recorded in detail, which will cause great trouble for readers. Although the experimental results seem to be rich, they are very poor in terms of evidence and discussion. In fact, if these mRNAs are all involved in their UV-B response, why the authors didn't find the proteins to prove it? It seems that these experimental results are only preliminary predictions, not actual results.

Response 2.1:

Thank you very much for your question and help us to improve the quality of our manuscript.

References has been modified.

Photosynthetic pigments including chlorophyll a (Chl a), chlorophyll b (Chl b), and total carotenoids were measured using previously published techniques [57].

[57] Wellburn, A.R. The Spectral Determination of Chlorophylls a and b, as well as Total Carotenoids, Using Various Solvents with Spectrophotometers of Different Resolution. Journal of Plant Physiology 1994, 5, 28-29.

Response 2.2:

Thank you very much for your question and help us to improve the quality of our manuscript.

the brand and model of GC-MS and chromatography column has been modified.

Terpenoids detection was analyzed using an Agilent 7890A gas chromatograph coupled to an Agilent 5975C Network Mass Selective Detector (MS, insert XL MSD with triple-axis detector).

Chromatographic column: HP-5MS column (30 m×0.25 mm×0.25 μm; J&W Scientific, Folsom, CA, USA);

Response 2.3:

Thank you very much for your question and help us to improve the quality of our manuscript.

Added experimental results

2.7 The dfr-miR156b-DfSPL3 module regulates the expression of DfGGPS1

MiR156 was first discovered in Arabidopsis thaliana. It consists of about 20 nucleotides and is structurally conserved [35]. It is involved in regulating plant growth and development by targeting the SPL family [36]. To verify the target relationship between dfr-miR156b and DfSPL3, we performed transient co-transformation technology in tobacco (Nicotiana benthamiana). Compared with the Pro35S :: GUS leaves, Pro35S:: DfSPL3-GUS exhibited the GUS phenotype revealed by histochemical staining, showed the DfSPL3 was fused the GUS gene. Compared with Pro35S:: DfSPL3-GUS leaves, the GUS staining was markedly decreased in leaves co-transformed with the strain mixture Pro35S :: DfSPL3-GUS and Pro35S :: dfr-miR156b (Figure 5A). This was considered to be additional evidence that dfr-miR156b could target DfSPL3. To confirm the results of these histochemical observations, RNA was extracted after two days of co-expression in tobacco, and the expression of DfSPL3 were analyzed by qRT-PCR. Compared with Pro35S :: DfSPL3-GUS leaves of DfSPL3 expression, the expression of DfSPL3 significantly declined in leaves co-transformed with the strain mixture Pro35S :: DfSPL3-GUS and Pro35S :: dfr-miR156b (Figure 5B). Taken together, these results show that dfr-miR156b targets DfSPL3.

The GTAC motif has been identified as the core binding site of SPLs [37,38]. Within a 2000-bp fragment of the DfGGPS1 gene upstream of the translation start codon, there are six GTAC motifs. Compared with the Pro35S :: GUS leaves, ProDfGGPS1:: GUS exhibited the GUS phenotype revealed by histochemical staining, showed the ProDfGGPS1 was activated GUS gene expression. Compared with ProDfGGPS1 :: GUS leaves, the GUS staining was markedly increased in leaves co-transformed with the strain mixture ProDfGGPS1:: GUS and 35S :: DfSPL3 (Figure 5C). This was considered to be additional evidence that DfSPL3 could interact with and activate the DfGGPS1. In order to confirm further the specific activation effect of DfSPL3 on the DfGGPS1 promoter, A Y1H verified that DfSPL3 could directly bind to the DfGGPS1 promoter (Figure 5D). These results indicated that DfSPL3 could interact with and activate the DfGGPS1.

Figure 5. A, β-Glucuronidase (GUS) phenotype observed by histochemical staining analysis of dfr-miR156b targets DfSPL3; B, Co-expression of the constructs containing Pro35S :: DfSPL3-GUS and Pro35S :: dfr-miR156b in tobacco leaves. Expression levels determined by qPCR were normalized to the expression levels of tobacco; C, β-Glucuronidase (GUS) phenotype observed by histochemical staining analysis of DfSPL3 binding to the DfGGPS1 promoter; D, Yeast one-hybrid analysis of DfSPL3 binding to the DfGGPS1 promoter, Interaction was determined on SD/-His/-Leu/-Trp (TDO) medium lacking leucine in the presence of 3AT. pGADT7-p53 and pHIS2-p53 were used as positive controls. pHIS2-empty and pHIS2-ProDfGGPS1 were used as negative controls; E, The simplified schematic networks of miRNA-mRNA networks for UV-B regulation of terpenoid metabolite synthesis.

References

  1. Yu, N.; Cai, W.J.; Wang, S.; Shan, C.M.; Wang, L.J.; Chen, X.Y. Temporal control of trichome distribution by microRNA156-targeted SPL genes in Arabidopsis thaliana. Plant Cell 2010, 22, 2322-2335.
  2. Wang, J.W.; Czech, B.; Weigel, D. miR156-regulated SPL transcription factors define an endogenous flowering pathway in Arabidopsis thaliana. Cell 2009, 138, 738-749.
  3. Birkenbihl, R.P.; Jach, G.; Saedler, H.; Huijser, P. Functional dissection of the plant-specific SBP-domain: overlap of the DNA-binding and nuclear localization domains. J Mol Biol 2005, 352, 585-596.
  4. Liang, X.; Nazarenus, T.J.; Stone, J.M. Identification of a consensus DNA-binding site for the Arabidopsis thaliana SBP domain transcription factor, AtSPL14, and binding kinetics by surface plasmon resonance. Biochemistry 2008, 47, 3645-3653.

Point 3: In this manualscript, the authors found that mRNA-miRNAs are involved in UV-B regulation of the synthesis of terpenoids in Fragrant Woodfern, but what is the content of terpenoids? What experimental results can the authors rely on to prove the results of Figure 3? At the same time, the resolution of Figure 3 is too poor and should be reproduced.

Response 3.1:

Thank you very much for your question and help us to improve the quality of our manuscript.

2.2. Terpenoid Content in Fragrant Woodfern Leaves Under UV-B Treatment

To investigate whether terpenoids in the leaves of Fragrant Woodfern are in-volved in the resistance response under UV-B stress, we treated Fragrant Woodfern with UV-B induction and collected metabolic samples of the leaves. We quantified the samples at different times of UV-B treatment by GC-MS method. As shown in the Supplementary Figure 3, we identified a total of nine terpenoids, namely Pentyl filicinate, β-Humulene, α Muurolene, Isolongifolol, Aristolochene, β-Maaliene, Drimenol, β-Cadinen, α-Patchoulene (Table S1). With the Compared with the control, the sum of the nine terpenes content did not change significantly (P > 0.05) at 0, 2 and 6 d of UV-B treatment, but the content of the nine terpenes was significantly higher (P < 0.05) at 4 d of UV-B treatment. So, we were selected Fragrant Woodfern leaves for mRNA and miRNA-seq analysis under 6 d treatment of UV-B.

Supplementary Figure 3. Effect of different UV-B radiation time on the contents of terpenoid in Fragrant Woodfern seedlings.

Supplementary Table 1 GC-MS detection of secondary metabolites in Fragrant Woodfern

No

molecular formula

CAS

description

1

C13H20O3

139-17-8

Pentyl filicinate

2

C15H24

116-04-1

β-Humulene

3

C15H24

31983-22-9

α-Muurolene

4

C15H26O

1139-17-9

Isolongifolol

5

C15H24

26620-71-3

Aristolochene

6

C15H24

72345-17-6

β-Maaliene

7

C15H26O

192724-27-9

Drimenol

8

C15H24

523-47-7

β-Cadinen

9

C15H24

560-32-7

α-Patchoulene

Response 3.2:

Thank you very much for your question and help us to improve the quality of our manuscript.

Figure 3 have been modified.

Figure 3 is a collection of kegg pathways (Photosynthesis, Flavonoid biosynthesis (ko00941), Plant hormone signal transduction, Phenylpropanoid biosynthesis, Glutathione metabolism, Starch and sucrose metabolism, Diterpenoid biosynthesis, Terpenoid backbone biosynthesis) involved in all DEGs (Table S3).

Figure 3. Related pathways of UV-B regulation of terpenoid synthesis in Fragrant Woodfern leaves.  Identified DEGs and Transcription factors detected in these pathways. The icons beside each gene name show the change in the gene differential expression level.

Point 4:. Are the results in Figure 5 evidence of protein? Only mRNA does not necessarily produce proteins. It is right that the synthetic pathway behind it is different. I don’t know why the authors have no considered this point.

Response 4.1:

Thank you very much for your question and help us to improve the quality of our manuscript.

Figure 4 wasn’t protein evidence. As shown in Figure 4, Regulatory network from the integrated analysis of miRNA-mRNA data. Co-expressed miRNA-mRNA interactions visualized as a network using Cytoscape (Table S4). Research shows that miRNAs negatively regulate target mRNA through translation repression or mRNA degradation. auxin response factor (ARFs) were the target gene of dfr-miRNA 160 (ARF17 and ARF18, targeted by miRNA160); transcription factor of SQUAMOSA promoter binding protein-like (SPL) gene was the target gene of dfr-miR156. Regulatory network from the integrated analysis of miRNA-mRNA data (Figure 4).

Figure 4. Regulatory network from the integrated analysis of miRNA-mRNA data. A, dfr-miRNA408-mRNAs of regulatory network; B, dfr-miRNA166a-mRNAs of regulatory network; C, dfr-miRNA160a-mRNAs of regulatory network; D, dfr-miRNA156b/c-mRNAs of regulatory network; A, dfr-miRNA171a/b/c-mRNAs of regulatory network; Co-expressed miRNA-mRNA interactions visualized as a network using Cytoscape. the circle represents mRNA, the Triangle represents miRNA, red represents up-regulation and blue represents down-regulation in network.

Point 5: In the discussion, it due to the insufficient experimental results, the reviewer believes that it is not detailed enough and is of limited help to readers. In fact, the mechanisms by which plants synthesize the secondary metabolites is quite complicated. The authors mentioned the cell death of fern leaves in the results. How can readers believe that the latter data are meaningful?

Response 5.1:

Thank you very much for your question and help us to improve the quality of our manuscript.

Added experimental results

2.7 The dfr-miR156b-DfSPL3 module regulates the expression of DfGGPS1

MiR156 was first discovered in Arabidopsis thaliana. It consists of about 20 nucleotides and is structurally conserved [35]. It is involved in regulating plant growth and development by targeting the SPL family [36]. To verify the target relationship between dfr-miR156b and DfSPL3, we performed transient co-transformation technology in tobacco (Nicotiana benthamiana). Compared with the Pro35S :: GUS leaves, Pro35S:: DfSPL3-GUS exhibited the GUS phenotype revealed by histochemical staining, showed the DfSPL3 was fused the GUS gene. Compared with Pro35S:: DfSPL3-GUS leaves, the GUS staining was markedly decreased in leaves co-transformed with the strain mixture Pro35S :: DfSPL3-GUS and Pro35S :: dfr-miR156b (Figure 5A). This was considered to be additional evidence that dfr-miR156b could target DfSPL3. To confirm the results of these histochemical observations, RNA was extracted after two days of co-expression in tobacco, and the expression of DfSPL3 were analyzed by qRT-PCR. Compared with Pro35S :: DfSPL3-GUS leaves of DfSPL3 expression, the expression of DfSPL3 significantly declined in leaves co-transformed with the strain mixture Pro35S :: DfSPL3-GUS and Pro35S :: dfr-miR156b (Figure 5B). Taken together, these results show that dfr-miR156b targets DfSPL3.

The GTAC motif has been identified as the core binding site of SPLs [37,38]. Within a 2000-bp fragment of the DfGGPS1 gene upstream of the translation start codon, there are six GTAC motifs. Compared with the Pro35S :: GUS leaves, ProDfGGPS1:: GUS exhibited the GUS phenotype revealed by histochemical staining, showed the ProDfGGPS1 was activated GUS gene expression. Compared with ProDfGGPS1 :: GUS leaves, the GUS staining was markedly increased in leaves co-transformed with the strain mixture ProDfGGPS1:: GUS and 35S :: DfSPL3 (Figure 5C). This was considered to be additional evidence that DfSPL3 could interact with and activate the DfGGPS1. In order to confirm further the specific activation effect of DfSPL3 on the DfGGPS1 promoter, A Y1H verified that DfSPL3 could directly bind to the DfGGPS1 promoter (Figure 5D). These results indicated that DfSPL3 could interact with and activate the DfGGPS1.

Figure 5. A, β-Glucuronidase (GUS) phenotype observed by histochemical staining analysis of dfr-miR156b targets DfSPL3; B, Co-expression of the constructs containing Pro35S :: DfSPL3-GUS and Pro35S :: dfr-miR156b in tobacco leaves. Expression levels determined by qPCR were normalized to the expression levels of tobacco; C, β-Glucuronidase (GUS) phenotype observed by histochemical staining analysis of DfSPL3 binding to the DfGGPS1 promoter; D, Yeast one-hybrid analysis of DfSPL3 binding to the DfGGPS1 promoter, Interaction was determined on SD/-His/-Leu/-Trp (TDO) medium lacking leucine in the presence of 3AT. pGADT7-p53 and pHIS2-p53 were used as positive controls. pHIS2-empty and pHIS2-ProDfGGPS1 were used as negative controls; E, The simplified schematic networks of miRNA-mRNA networks for UV-B regulation of terpenoid metabolite synthesis.

Response 5.2:

Cell death indicated as a loss of plasma membrane integrity [22]. In recent years, with the destruction of the earth's ozone layer, the physiological and ecological studies of UV-B radiation on plants have received increasing attention from scholars [39]. UV-B radiation can cause imbalance of ROS metabolism and pro-duce a large amount of superoxide ions, which can intensify the peroxidation of membrane lipids and eventually cause damage to membrane structure and normal physiological functions of plants [40]. The main enzymatic scavenging systems for ROS scavenging in plant cells are SOD, CAT and POD [41]. When plants are stressed, the activities of these antioxidant enzymes change accordingly and are involved in the regulation of ROS metabolism to protect plants from environmental stresses [42].

UV-B radiation can cause imbalance of ROS metabolism and produce a large amount of superoxide ions, which intensifies the peroxidation of membrane lipids, eventually causing damage to the membrane structure and damage to the normal physiological functions of plants. We examined cell death.

References

  1. Andrady, A.L.; Aucamp, P.J.; Austin, A.T.; Bais, A.F.; Ballaré, C.L.; Björn, L.O.; Bornman, J.F.; Caldwell, M.; Cullen, A.P.; Erickson, D.J.; et al. Environmental effects of ozone depletion and its interactions with climate change: progress report, 2011. Photochem Photobiol Sci 2012, 11, 13-27.
  2. Yu, N.; Cai, W.J.; Wang, S.; Shan, C.M.; Wang, L.J.; Chen, X.Y. Temporal control of trichome distribution by microRNA156-targeted SPL genes in Arabidopsis thaliana. Plant Cell 2010, 22, 2322-2335.
  3. Wang, J.W.; Czech, B.; Weigel, D. miR156-regulated SPL transcription factors define an endogenous flowering pathway in Arabidopsis thaliana. Cell 2009, 138, 738-749.
  4. Birkenbihl, R.P.; Jach, G.; Saedler, H.; Huijser, P. Functional dissection of the plant-specific SBP-domain: overlap of the DNA-binding and nuclear localization domains. J Mol Biol 2005, 352, 585-596.
  5. Liang, X.; Nazarenus, T.J.; Stone, J.M. Identification of a consensus DNA-binding site for the Arabidopsis thaliana SBP domain transcription factor, AtSPL14, and binding kinetics by surface plasmon resonance. Biochemistry 2008, 47, 3645-3653.
  6. Zhang, W.J.; Björn, L.O. The effect of ultraviolet radiation on the accumulation of medicinal compounds in plants. Fitoterapia 2009, 80, 207-218.
  7. Takshak, S.; Agrawal, S.B. Secondary metabolites and phenylpropanoid pathway enzymes as influenced under supplemental ultraviolet-B radiation in Withania somnifera Dunal, an indigenous medicinal plant. J Photochem Photobiol B 2014, 140, 332-343.
  8. Sun, N.; Song, T.; Ma, Z.; Dong, L.; Zhan, L.; Xing, Y.; Liu, J.; Song, J.; Wang, S.; Cai, H. Overexpression of MsSiR enhances alkali tolerance in alfalfa (Medicago sativa L.) by increasing the glutathione content. Plant Physiol Biochem 2020, 154, 538-546.
  9. Bao, Y.; Yang, N.; Meng, J.; Wang, D.; Fu, L.; Wang, J.; Cang, J. Adaptability of winter wheat Dongnongdongmai 1 (Triticum aestivum L.) to overwintering in alpine regions. Plant Biol (Stuttg) 2021, 23, 445-455.

Reviewer 2 Report

Reviewer’s Comment of ijms-1689272

Photo-regulation of plant hormone signal transduction has attracted great attentions in the recent years. In this study, the authors found that UV-B could regulate the terpenoids synthesis of molecular regulatory mechanism in Fragrant Woodfern by mRNA and small-RNA libraries analysis. Nine terpenoids were identified. The results showed that the target genes were mainly enriched in diterpene biosynthesis, terpenoid backbone biosynthesis, plant hormone signal transduction, MEP pathway and MVA pathway, in which miR156 and miR160 regulate these pathways by targeting DfSPL and DfARF, respectively. This work provided useful information that mRNA-miRNAs were involved in UV-B regulates terpenoids synthesis in Fragrant Woodfern. There are several issues need to be addressed before further consideration:

  1. The abstract is a bit redundant and not well-organized; the authors please re-organize the abstract to simplify it.
  2. For a better comparison, the GC-MS results with molecular structures of the terpenoids need to be summarized and presented in supporting information.
  3. In the maintext, “β Humulene, α Muurolene, β Maaliene, β Cadinen, α Patchoulene” should be “β-Humulene, α-Muurolene, β-Maaliene, β-Cadinen, α-Patchoulene”.
  4. In Table 2, “Tota reads” should be “Total reads”.
  5. In Figure 3, the text size is a bit small; the authors please enlarge the text and make them clearer.
  6. In Figure 5. Regulatory network from the integrated analysis of miRNA-mRNA data, quality of the pictures need to be improved to make them clearer.
  7. In supporting information, there are some format/spelling problems: e.g. “Protein CANDIDATE G-PROTEIN COUPLED RECEPTOR 7”; “Phot ystem I subunit O”; “Lys omal beta glucidase”, and so on. The authors please carefully check and correct these places.
  8. There are some format problems in the reference part, e.g. ref. 1. “…Int J Mol Sci 2020, 21.”; ref.3. “…Molecules 2018, 23.”; ref.44. “…Molecules 2019, 24.”; ref. 51. “…Int J Mol Sci 2019, 20.”, and so on. The authors please carefully check and correct the reference part.

Author Response

Response to Reviewer 2 Comments

Point 1: The abstract is a bit redundant and not well-organized; the authors please re-organize the abstract to simplify it.

Response 1:

Thank you very much for your question and help us to improve the quality of our manuscript.

Abstract has been modified.

Abstract: Fragrant Woodfern (Dryopteris fragrans) is a medicinal plant rich in terpenoids. Ultra-violet-B (UV-B) light could increase concentration of terpenoids. The aim of this study was to UV-B regulates the terpenoids synthesis of molecular regulatory mechanism in Fragrant Woodfern. In this study, compare with the control group, the content of the terpenes was significantly higher at Fragrant Woodfern leaves under UV-B treatment of 4d. In order to identify UV-B regulatory terpenoid metabolic of mechanism in Fragrant Woodfern. we examined the mRNAs and small RNAs in Fragrant Woodfern leaves under UV-B treatment. mRNA and miRNA-seq identified 4533 DEGs and 17 DEMs in control group compared with Fragrant Woodfern leaves under UV-B treatment of 4 d. mRNA-miRNA analysis identified miRNA target gene pairs consisting of 8 DEMs and 115 miRNAs. The target genes were subjected to GO, and KEGG analyses. The results showed that the target genes were mainly enriched in diterpene biosynthesis, terpenoid backbone biosynthesis, plant hormone signal trans-duction, MEP pathway, and MVA pathway, in which miR156 and miR160 regulate these pathways by targeting DfSPL and DfARF, respectively. The mRNA and miRNA datasets identified a subset of candidate genes. It provides the theoretical basis that UV-B regulates the terpenoids synthesis of molecular regulatory mechanism in Fragrant Woodfern.

Point 2: For a better comparison, the GC-MS results with molecular structures of the terpenoids need to be summarized and presented in supporting information.

Response 1:

Thank you very much for your question and help us to improve the quality of our manuscript.

molecular structures of the terpenoids have been modified.

Supplementary Table 1 GC-MS detection of secondary metabolites in Fragrant Woodfern

No

molecular formula

CAS

description

1

C13H20O3

139-17-8

Pentyl filicinate

2

C15H24

116-04-1

β-Humulene

3

C15H24

31983-22-9

α-Muurolene

4

C15H26O

1139-17-9

Isolongifolol

5

C15H24

26620-71-3

Aristolochene

6

C15H24

72345-17-6

β-Maaliene

7

C15H26O

192724-27-9

Drimenol

8

C15H24

523-47-7

β-Cadinen

9

C15H24

560-32-7

α-Patchoulene

Point 3: In the maintext, “β Humulene, α Muurolene, β Maaliene, β Cadinen, α Patchoulene” should be “β-Humulene, α-Muurolene, β-Maaliene, β-Cadinen, α-Patchoulene”.

Response 3:

Thank you very much for your question and help us to improve the quality of our manuscript.

“β-Humulene, α-Muurolene, β-Maaliene, β-Cadinen, α-Patchoulene” have been modified.

Point 4: In Table 2, “Tota reads” should be “Total reads”.

Response 4:

Thank you very much for your question and help us to improve the quality of our manuscript.

“Total reads” have been modified.

Table 2. Statistic analysis of clean reads for small RNA sequencing in Fragrant Woodfern leaves.

Sample

Total reads

Clean reads

Mapped sRNA

Known miRNA

Novel miRNA

Total miRNA

CK1

15490143 (100.00%)

14585008 (94.16%)

6307041 (58.03%)

4

72

76

CK2

12790994 (100.00%)

12417161 (97.08%)

4561120 (51.72%)

6

71

77

CK3

15716150 (100.00%)

15055676 (95.80%)

6417540 (67.93%)

5

72

77

UV1

17010191 (100.00%)

16386263 (96.33%)

6917716 (61.08%)

6

77

83

UV2

13075551 (100.00%)

12297545 (94.05%)

4580695 (57.97%)

7

79

86

UV3

11728662 (100.00%)

11077797 (94.45%)

4023081 (54.21%)

6

78

84

Sum

85811691

81819450

Point 5: In Figure 3, the text size is a bit small; the authors please enlarge the text and make them clearer.

Response 5:

Thank you very much for your question and help us to improve the quality of our manuscript.

Figure 3 have been modified.

Figure 3. Related pathways of UV-B regulation of terpenoid synthesis in Fragrant Woodfern leaves.  Identified DEGs and Transcription factors detected in these pathways. The icons beside each gene name show the change in the gene differential expression level.

Point 6: In Figure 5. Regulatory network from the integrated analysis of miRNA-mRNA data, quality of the pictures need to be improved to make them clearer.

Response 5:

Thank you very much for your question and help us to improve the quality of our manuscript.

Figure 4 have been modified.

Figure 4. Regulatory network from the integrated analysis of miRNA-mRNA data. A, dfr-miRNA408-mRNAs of regulatory network; B, dfr-miRNA166a-mRNAs of regulatory network; C, dfr-miRNA160a-mRNAs of regulatory network; D, dfr-miRNA156b/c-mRNAs of regulatory network; A, dfr-miRNA171a/b/c-mRNAs of regulatory network; Co-expressed miRNA-mRNA interactions visualized as a network using Cytoscape. the circle represents mRNA, the Triangle represents miRNA, red represents up-regulation and blue represents down-regulation in network.

Point 7: In supporting information, there are some format/spelling problems: e.g. “Protein CANDIDATE G-PROTEIN COUPLED RECEPTOR 7”; “Phot ystem I subunit O”; “Lys omal beta glucidase”, and so on. The authors please carefully check and correct these places.

Response 5:

Thank you very much for your question and help us to improve the quality of our manuscript.

supporting information have been modified.

Supplementary Table 4. Regulatory network from the integrated analysis of miRNA-mRNA data

miRNA ID

Fold change

mRNA ID

Fold change

Description

dfr-miR156b

-8.5365

Cluster-10273.0

-1.7485

Squama promoter-binding-like protein 3

Cluster-123.29223

3.5728

Arogenate dehydratase 3

Cluster-123.29409

2.9169

Guanylate kinase 1

Cluster-23659.0

-2.8911

L-type lectin-domain containing receptor kinase

Cluster-123.13809

-2.8843

Pentatricopeptide repeat-containing protein

Cluster-123.19695

-1.9519

CBS domain-containing protein CBSX3

Cluster-123.8725

-1.6135

Putative glutathione peroxidase 7

Cluster-123.5408

1.8182

Light-inducible protein CPRF2

Cluster-123.1038

5.4348

Transcription factor MYB20

dfr-miR156c

-3.2

Cluster-10273.0

-1.7485

Squama promoter-binding-like protein 3

Cluster-123.19313

1.7138

Probable serine/threonine-protein kinase SIS8

Cluster-123.20543

-2.7227

Transcription repressor KAN1

Cluster-123.6557

-1.6352

Pentatricopeptide repeat-containing protein

Cluster-123.7903

1.7956

G protein-coupled receptor 7

Cluster-123.16999

2.211

Probable protein phosphatase 2C

Cluster-123.14057

2.6173

Histone H3.3-like type 2

Cluster-123.25881

2.6414

Tuberculostearic acid methyltransferase

Cluster-123.20114

2.7506

B2 protein  

Cluster-123.29223

3.5728

Arogenate dehydratase 3

Cluster-123.28610

-2.091

Pentatricopeptide repeat-containing protein

Cluster-123.23487

-1.6409

Molybdenum cofactor sulfurase

Cluster-123.22583

2.0388

IAA-amino acid hydrolase ILR1-like 3

Cluster-123.26549

-3.3717

Long-chain-alcohol oxidase

Cluster-123.22415

-3.1846

Glucan endo-1,3-beta-glucidase 12

Cluster-23659.0

-2.8911

L-type lectin-domain containing receptor kinase

Cluster-123.18173

-2.621

Fructe-1,6-bisphphatase

Cluster-123.17902

-1.9966

2-methylene-furan-3-one reductase

Cluster-123.19695

-1.9519

CBS domain-containing protein CBSX3

Cluster-123.20747

2.0049

Homeobox-DDT domain protein RLT1

Cluster-123.22763

2.3964

Linamarin synthase 2

Cluster-123.20103

2.6536

UDP-glycosyltransferase

Cluster-123.24349

2.7405

O-fucosyltransferase 19

Cluster-123.4022

2.9833

Transcription factor KUA1

Cluster-123.23959

3.1297

Disease resistance protein TAO1

Cluster-12165.0

3.3156

Histone H3.2

Cluster-123.14806

3.4865

Scarecrow-like protein 14

dfr-miR160a

-8.0234

Cluster-123.3474

2.061

Auxin response factor 18

Cluster-123.4068

2.054

Auxin response factor 16

Cluster-123.2933

2.3179

Probable leucine-rich repeat receptor-like protein kinase

Cluster-123.3200

2.5757

Beta-glucosidase 18

Cluster-123.5888

5.5054

Mitochondrial carrier protein CoAc1

Cluster-123.16555

-3.8488

Chalcone synthase

Cluster-123.17914

-3.2321

photosystem I subunit PsaO

Cluster-123.14287

-2.3081

RHOMBOID-like protein 9

Cluster-123.20187

2.044

Membrane-anchored ubiquitin-fold protein 3

Cluster-123.18740

2.306

Lysosomal beta glucosidase

Cluster-123.4405

2.5092

Cytokinin hydroxylase  

Cluster-123.28431

2.8925

G-type lectin S-receptor-like serine/threonine-protein kinase

dfr-miR166a

-8.0234

Cluster-123.14967

2.2091

Disease resistance protein L6

Cluster-123.10655

-2.4092

Calcium-transporting ATPase 8

Cluster-123.16944

-2.9765

UDP-glycosyltransferase

Cluster-123.9847

4.9293

Putative methyltransferase

dfr-miR171a

-19.5771

Cluster-123.10974

-2.3492

Scavenger receptor class B member 1

Cluster-123.16151

-2.1084

Dormancy-associated protein 1

Cluster-123.19430

2.969

Scarecrow-like protein 14

Cluster-24393.0

-4.3188

Bifunctional nuclease 1

Cluster-123.19430

2.969

Scarecrow-like protein 14

Cluster-123.10411

3.3118

UDP-galactose/UDP-glucose transporter 7

Cluster-123.27995

5.4762

Mitogen-activated protein kinase kinase kinase A

dfr-miR171b

0.2481

Cluster-123.19430

2.969

Scarecrow-like protein 14  

Cluster-123.10974

-2.3492

Scavenger receptor class B member 1  

Cluster-123.29383

2.1689

Zinc finger CCCH domain-containing protein 24  

Cluster-123.16739

-3.6975

Photosystem II core complex proteins psbY

Cluster-123.16151

-2.1084

Dormancy-associated protein 1  

Cluster-123.11662

1.7213

Serine/threonine-protein kinase STY13  

Cluster-123.18601

2.2211

Serine/threonine-protein kinase PCRK2  

Cluster-26561.0

2.3745

Probable long-chain-alcohol O-fatty-acyltransferase 5  

Cluster-123.21047

3.8836

Unsaturated rhamnogalacturonyl hydrolase YteR

Cluster-123.20658

5.4116

Probable L-type lectin-domain containing receptor kinase S.7  

Cluster-123.27185

-2.6199

Pentatricopeptide repeat-containing protein DOT4,

Cluster-123.12783

2.0341

Probable serine/threonine-protein kinase PIX7  

Cluster-123.18228

2.0593

C2 and GRAM domain-containing protein At5g50170  

Cluster-123.23405

2.842

RHOMBOID-like protein 2  

Cluster-123.19430

2.969

Scarecrow-like protein 14  

Cluster-123.23589

3.5155

Probable sucrose-phosphate synthase 2

Cluster-123.30629

4.6644

Calcium-dependent protein kinase 17  

Cluster-33680.0

4.694

Probable L-type lectin-domain containing receptor kinase

dfr-miR171c

Cluster-123.18266

2.255

Glutathione S-transferase F10  

Cluster-123.12337

3.2048

Endoplasmic reticulum oxidoreductin-1

Cluster-123.18771

-5.8455

40S ribosomal protein S15

Cluster-123.8398

-2.2953

Pentatricopeptide repeat-containing protein

Cluster-123.9159

2.3147

FAS1 domain-containing protein

Cluster-123.21840

2.5269

Aconitate hydratase

dfr-miR408

-2.72096

Cluster-123.10871

6.1936

Leucine-rich repeat receptor-like serine/threonine-protein kinase

Cluster-123.14233

-2.0905

Ferredoxin-1

Cluster-123.15627

1.622

Probable polyamine transporter

Cluster-123.4444

3.7865

Light-sensor Protein kinase  

Cluster-22443.0

-2.8687

Leucine-rich repeat receptor-like serine/threonine-protein kinase

Cluster-123.16299

1.6082

NADP-dependent glyceraldehyde-3-phphate dehydrogenase  

Point 8: There are some format problems in the reference part, e.g. ref. 1. “…Int J Mol Sci 2020, 21.”; ref.3. “…Molecules 2018, 23.”; ref.44. “…Molecules 2019, 24.”; ref. 51. “…Int J Mol Sci 2019, 20.”, and so on. The authors please carefully check and correct the reference part.

Response 8:

Thank you very much for your question and help us to improve the quality of our manuscript.

References have been modified.

  1. Chen, L.; Zhang, D.; Song, C.; Wang, H.; Tang, X.; Chang, Y. Transcriptomic Analysis and Specific Expression of Transcription Factor Genes in the Root and Sporophyll of Dryopteris fragrans (L.) Schott. Int J Mol Sci 2020, 21, 7296.
  2. Zhang, T.; Wang, L.; Duan, D.H.; Zhang, Y.H.; Huang, S.X.; Chang, Y. Cytotoxicity-Guided Isolation of Two New Phenolic Derivatives from Dryopteris fragrans (L.) Schott. Molecules 2018, 23, 1652.
  3. Xu, Y.; Zhu, C.; Xu, C.; Sun, J.; Grierson, D.; Zhang, B.; Chen, K. Integration of Metabolite Profiling and Transcriptome Analysis Reveals Genes Related to Volatile Terpenoid Metabolism in Finger Citron (C. medica var. sarcodactylis). Molecules 2019, 24, 2564.
  4. Wang, X.; Chen, X.; Zhong, L.; Zhou, X.; Tang, Y.; Liu, Y.; Li, J.; Zheng, H.; Zhan, R.; Chen, L. PatJAZ6 Acts as a Repressor Regulating JA-Induced Biosynthesis of Patchouli Alcohol in Pogostemon Cablin. Int J Mol Sci 2019, 20, 6038.

Round 2

Reviewer 1 Report

  1. Although the authors removed the figure 5 and changed it to Figure 3, and added the performance of some proteins in the new version of manuscript. But for figure 3, the reviewer believes that only these data cannot support the results of figure 3. It is recommended that the authors revise it and the established the mechanisms are fine.
  2. For the amount of GC-MS, the authors still does not have a data to show their experimental quantitative results. Secondly, the authors also still has no reasonable explanations for the experimental results that UVB causes cell death. Can the dead cells have different internal chemicals comparable (secondary metabolites) to compare with the normal living cells?

My suggestion is still major revision.

Author Response

Response to Reviewer 1 Comments

Point 1: Although the authors removed the figure 5 and changed it to Figure 3, and added the performance of some proteins in the new version of manuscript. But for figure 3, the reviewer believes that only these data cannot support the results of figure 3. It is recommended that the authors revise it and the established the mechanisms are fine.

Response 1:

Despite their large structural diversity, all terpenoids are synthesized from the universal five carbon precursors, isopentenyl diphosphate (IPP) and its allylic isomer dimethylallyl diphosphate (DMAPP). In plants, these precursors are derived from two alternate and independent biosynthetic pathways localized in different subcellular compartments. While mevalonic-acid (MVA) pathway gives rise to IPP starting from acetyl-CoA in the cytosol, 2-C-methy-lerythritol 4-phosphate (MEP) pathway leads to the formation of IPP and DMAPP from pyruvate and glyceraldehyde-3-phosphate in the plastids. acetyl-CoA is produced by the TCA cycle. Pyruvate is produced by the glycolytic pathway.

The purpose of the expression of Figure 3 is a synthetic pathway of a terpenoid, wherein the substrate and product of the above, can be found on the KEEG. We consolidate the glycolytic pathway, TCA cycle, MVA pathway and MEP pathway into Figure 3.

Point 2: For the amount of GC-MS, the authors still does not have a data to show their experimental quantitative results. Secondly, the authors also still has no reasonable explanations for the experimental results that UVB causes cell death. Can the dead cells have different internal chemicals comparable (secondary metabolites) to compare with the normal living cells?

Response 2.1:

Put 0.1g of Dryopteris fragrant leaves and 1ml of ethyl acetate standard solution into a 1.5ml EP tube. which contained 8.64 µg/ml nonyl acetate as internal standard.

2.2. Terpenoid Content in Fragrant Woodfern Leaves Under UV-B Treatment

To investigate whether terpenoids in the leaves of Fragrant Woodfern are involved in the resistance response under UV-B stress, we treated Fragrant Woodfern with UV-B induction and collected metabolic samples of the leaves. We quantified the samples at different times of UV-B treatment by GC-MS method. As shown in the Supplementary Figure 3, we identified a total of nine terpenoids, namely Pentyl filicinate, β-Humulene, α Muurolene, Isolongifolol, Aristolochene, β-Maaliene, Drimenol, β-Cadinen, α-Patchoulene (Table S1). With the Compared with the control, the sum of the nine terpenes content did not change significantly (P > 0.05) at 0, 2 and 6 d of UV-B treatment, but the content of the nine terpenes was significantly higher (P < 0.05) at 4 d of UV-B treatment. So, we were selected Fragrant Woodfern leaves for mRNA and miRNA-seq analysis under 6 d treatment of UV-B.

Supplementary Figure 3. Effect of different UV-B radiation time on the contents of terpenoid in Fragrant Woodfern seedlings.

CK

UV

Supplementary Table 1 GC-MS detection of secondary metabolites in Fragrant Woodfern

No

molecular formula

CAS

description

1

C13H20O3

139-17-8

Pentyl filicinate

2

C15H24

116-04-1

β-Humulene

3

C15H24

31983-22-9

α-Muurolene

4

C15H26O

1139-17-9

Isolongifolol

5

C15H24

26620-71-3

Aristolochene

6

C15H24

72345-17-6

β-Maaliene

7

C15H26O

192724-27-9

Drimenol

8

C15H24

523-47-7

β-Cadinen

9

C15H24

560-32-7

α-Patchoulene

Response 2.2:

Cell death indicators represent loss of plasma membrane integrity. UV-B radiation can cause imbalance of ROS metabolism and produce a large amount of superoxide ions, which intensifies the peroxidation of membrane lipids, eventually causing damage to the membrane structure and damage to the normal physiological functions of plants.

The aim of this study was to UV-B regulates the terpenoids synthesis of molecular regulatory mechanism in Fragrant Woodfern. the dead cells have different internal chemicals comparable (secondary metabolites) to compare with the normal living cells as the next research in our laboratory.

Reviewer 2 Report

The comments were successfully addressed by the authors in the revised manuscript of ijms-1689272.

Author Response

Thank you very much for your question and help us to improve the quality of our manuscript.

Round 3

Reviewer 1 Report

With regarding my comments, the authors haves not made any improvements, and I maintain my suggestion.

(The secondary metabolites has no quantity)

My suggestion is still major revision.